JCB — Journal of Cell Biology

# The pathway of unconventional protein secretion involves CUPS and a modified trans-Golgi network

Amy J. Curwin[1]* , Kazuo Kurokawa[4]* , Gonzalo Bigliani[1] , Nathalie Brouwers[1] , Akihiko Nakano[4] , and Vivek Malhotra[1,2,3]

Compartment for unconventional protein secretion (CUPS), a compartment for secretion of signal sequence–lacking proteins, forms through COPI-independent extraction of membranes from early Golgi cisternae, lacks Golgi-specific glycosyltransferases, and requires phosphatidylinositol 4-phosphate (PI4P) for biogenesis, as well as phosphatidylinositol 3-phosphate for stability. Our findings demonstrate that Drs2, a PI4P effector from the trans-Golgi network (TGN), is essential for CUPS formation, specifically through its interaction with Rcy1, and Rcy1 is crucial for the unconventional secretion. Using 4D super-resolution confocal live imaging microscopy, we observed that CUPS interact with a modified TGN that contains Drs2 in addition to proteins Tlg2 and Snc2, which are necessary for membrane fusion. Notably, while CUPS remain stable, the modified TGN undergoes remodeling during the later stages of unconventional secretion. In summary, we suggest that CUPS and the modified TGN, without the function of COPII and COPI, participate in collecting and sorting unconventionally secreted proteins, reflecting the role of Golgi membranes in receiving cargo from the ER during conventional secretion.

## Introduction

How proteins that cannot enter the ER–Golgi pathway of secretion are released into the extracellular space remains a fascinating question. This issue is particularly important because cells secrete proteins such as fibroblast growth factor (FGF) 2, interleukin (IL)-1β, acyl-CoA binding protein (Acb1/diazepam binding inhibitor), superoxide dismutase (SOD) 1, and tissue transglutaminase, all of which play significant physiological roles in the extracellular environment, especially under conditions of stress.

Current models for the secretion of this class of cytoplasmic proteins propose several mechanisms: the involvement of an intracellular compartment, such as compartments for unconventional protein secretion (CUPS), for proteins such as Acb1 and SOD1, along with many other antioxidants; secretory endosomes or lysosomes for fatty acid binding protein 4; and direct translocation across the plasma membrane for FGF2 (Bruns et al., 2011; Cruz-Garcia et al., 2020; Lolicato et al., 2022; Padmanabhan and Manjithaya, 2023; Schäfer et al., 2004; Villeneuve et al., 2018). IL-1β is reported to employ multiple export pathways, including translocation directly across the plasma membrane via a pore formed by gasdermin D, translocation through the conventional cargo receptor protein TMED10 into the ER–Golgi intermediate compartment (ERGIC) before being released from cells, and secretion via pyroptosis (Chiritoiu et al., 2019; He et al., 2015; Liu et al., 2016).

We have focused on the pathway for the secretion of Acb1 and SOD1 from yeast, which requires several essential factors: (1) their secretion is triggered by carbon and nitrogen starvation, as well as growth in potassium acetate; (2) the intracellular production of ROS; (3) the involvement of a peripherally localized Golgi/ER exit site protein called Grh1 (GORASPs in metazoans); and (4) a compartment known as CUPS (Bruns et al., 2011; Cruz-Garcia et al., 2014, 2020; Curwin et al., 2016; Kinseth et al., 2007). It is of note that the release of IL-1β is also dependent on GORASPs in LPS-activated macrophages (Chiritoiu et al., 2019). GORASPs are currently the only proteins identified that facilitate various types of unconventional secretion across different organisms throughout evolution (Chiritoiu-Butnaru et al., 2022).

In yeast, CUPS are characterized by the presence of the single GORASP ortholog, Grh1 (Bruns et al., 2011). CUPS form independently of COPI and COPII proteins but require the activity of the phosphatidylinositol (PI) 4-kinase, Pik1, located in late Golgi membranes or the trans-Golgi network (TGN) (Cruz-Garcia et al., 2014). During starvation, correlating with the timing of unconventional secretion, CUPS undergo "maturation," acquiring large enveloping membranes in a process that necessitates the function of the PI3-kinase Vps34 and a subset of ESCRT proteins. In the absence of Vps34 or ESCRT complexes I, II, or III, CUPS initially form but later fragment. The major subunit of

---

[1]Centre for Genomic Regulation, The Barcelona Institute of Science and Technology, Barcelona, Spain;   [2]Universitat Pompeu Fabra (UPF), Barcelona, Spain;   [3]ICREA, Barcelona, Spain;   [4]Live Cell Super-Resolution Imaging Research Team, RIKEN Center for Advanced Photonics, Wako, Japan.

*A.J. Curwin and K. Kurokawa contributed equally to this paper.   Correspondence to Vivek Malhotra: vivek.malhotra@crg.eu.

ESCRT-III, Snf7, also transiently localizes to the CUPS (Curwin et al., 2016).

We now report that the TGN-localized and functionally active aminophospholipid flippase Drs2, a phosphatidylinositol 4-phosphate (PI4P) effector, is essential for CUPS biogenesis. This requirement is dependent on the Drs2 binding partner Rcy1. Our data reveal that during unconventional secretion, a modified TGN containing Drs2, along with proteins required for membrane fusion—Tlg2 (t-SNARE) and Snc2 (v-SNARE)—transiently interacts with Grh1-containing CUPS. 4D imaging of cells using super-resolution confocal live imaging microscopy (SCLIM) demonstrated that CUPS and the modified TGN make numerous transient contacts, which are likely functionally significant for unconventional protein secretion. The following section describes our findings in detail.

## Results

### PI4P effector Drs2 is necessary for CUPS biogenesis

PI4P is produced at the TGN by the PI4-kinase Pik1 and is essential for proper Golgi function through various PI4P-dependent pathways (Graham and Burd, 2011; Walch-Solimena and Novick, 1999). By the use of a temperature-sensitive allele, it has been demonstrated that Pik1 is required for efficient CUPS biogenesis. However, a PI4P fluorescent sensor does not localize to the Golgi; instead, it is diffusely dispersed in the cytoplasm under starvation conditions, suggesting a reduction in Golgi-specific PI4P levels (Cruz-Garcia et al., 2014).

This finding aligns with published research indicating that glucose starvation leads to a rapid decrease in Golgi PI4P levels due to the relocalization of the enzymes Pik1 and the PI4P Sac1 (Demmel et al., 2008; Faulhammer et al., 2007). What role do the late Golgi membranes and Pik1 play in the overall process of CUPS formation, and what are the effectors of PI4P in this pathway?

The multispanning transmembrane protein Drs2 is a lipid flippase and PI4P effector localized to TGN membranes. We examined the localization of genomically expressed Grh1-2xmCherry and Drs2-3xGFP during growth and throughout the starvation time course using live-cell microscopy. Under growth conditions, Drs2 labeled four to six punctate structures per cell that were often apposed to, but not colocalized with, Grh1 (Fig. 1 A). Upon starvation, Grh1 relocalized to one to three larger foci, which we have previously identified as CUPS. Curiously, Drs2, a transmembrane protein, also relocalized to one to three larger foci per cell in addition to a diffuse vesicular staining in the cytoplasm (Fig. 1 A).

We observed transient colocalization of the foci of Grh1 and Drs2, particularly early in starvation (26% of cells). The rate of transient colocalization decreased throughout the starvation period (10% of cells). This decline may reflect the overall reduction in the number of Drs2 compartments as starvation progressed. The average number of Drs2 structures per cell decreased from 2.3 in the first 45 min to 1.5 in the last 30 min of starvation, while the average percentage of cells with no Drs2 structures increased from 7.9% to 13.8% during the same time-frame (Fig. 1).

Next, we investigated whether Drs2 contributes to the process of CUPS formation. We examined the localization of Grh1-2xGFP in cells lacking Drs2 and found that Grh1 localized to numerous smaller structures instead of the typical one to three large puncta associated with CUPS (Fig. 1 B). Even after 2.5–3 h of starvation, Grh1-2xGFP remained in smaller structures rather than the CUPS we had previously defined. Thus, while Drs2 and Grh1 show considerable colocalization early in starvation, this interaction declines with prolonged starvation. We also confirmed that the CUPS defect observed upon loss of Drs2 is dependent on its phospholipid flippase activity. The D560N mutation is known to completely abolish the enzymatic activity of Drs2, which was confirmed in in vitro assays (Chen et al., 1999). Cells lacking DRS2 are cold-sensitive and prone to suppression (e.g., they can easily revert to previous phenotypes). We observed that the CUPS defect in drs2Δ cells can also easily revert, with freshly generated strains displaying stronger CUPS defects. To test the dependence on flippase activity, we set up a system to transiently turn off Drs2 expression to avoid suppression effects while also expressing an empty vector, wild-type (WT) Drs2, or flippase-inactive Drs2. We established this system in cells also expressing Grh1-2xmCherry and observed rescue by WT Drs2, but not by the flippase-inactive mutant or the empty vector, in cells where Drs2 expression had been previously turned off (Fig. S1). Therefore, the observed CUPS defect is indeed dependent on the flippase activity of Drs2.

### Drs2 functions specifically with Rcy1 in CUPS formation

Drs2 functions at the TGN to flip primarily phosphatidylserine but also phosphatidylethanolamine, maintaining phospholipid asymmetry and driving vesicle formation (Chen et al., 1999; Gall et al., 2002; Hua et al., 2002; Liu et al., 2008; Natarajan et al., 2004). The C-terminal domain of Drs2 has an autoinhibitory function that is relieved upon binding to PI4P. The interaction of Drs2 with the Arf-GEF Gea2 and the Arf-like GTPase Arl1 is also critical for regulating multiple clathrin-dependent anterograde pathways (Bai et al., 2019; Hankins et al., 2015; Natarajan et al., 2009; Timcenko et al., 2019; Tsai et al., 2013; Zhou et al., 2013). Since clathrin and the Drs2 PI4P sensor dissociate from the TGN upon starvation, we did not expect these components to play a role in CUPS formation. Nevertheless, we tested mutant strains lacking the functions of Gea2, Arl1, and clathrin (clathrin heavy chain or adaptor proteins), and their loss revealed no effect on CUPS formation (Fig. S2).

Drs2, through its interaction with Rcy1, also regulates a retrograde pathway required for the recycling of the exocytic v-SNARE Snc1 to the TGN. Rcy1 is an F-box–containing protein that binds Drs2 via its C-terminal domain, in a region proximal to the PI4P binding site, which partially overlaps with the Gea2 binding site (Furuta et al., 2007; Hanamatsu et al., 2014). This function of Rcy1 in retrograde trafficking is independent of the cullin-Ub–conjugating E2 ligase (Cdc34) pathway (Galan et al., 2001). Deletion of RCY1 resulted in highly vesiculated Grh1-positive structures, similar to those observed upon loss of Drs2. These data indicate that the Drs2-Rcy1 pathway is specifically required for CUPS formation (Fig. 2).

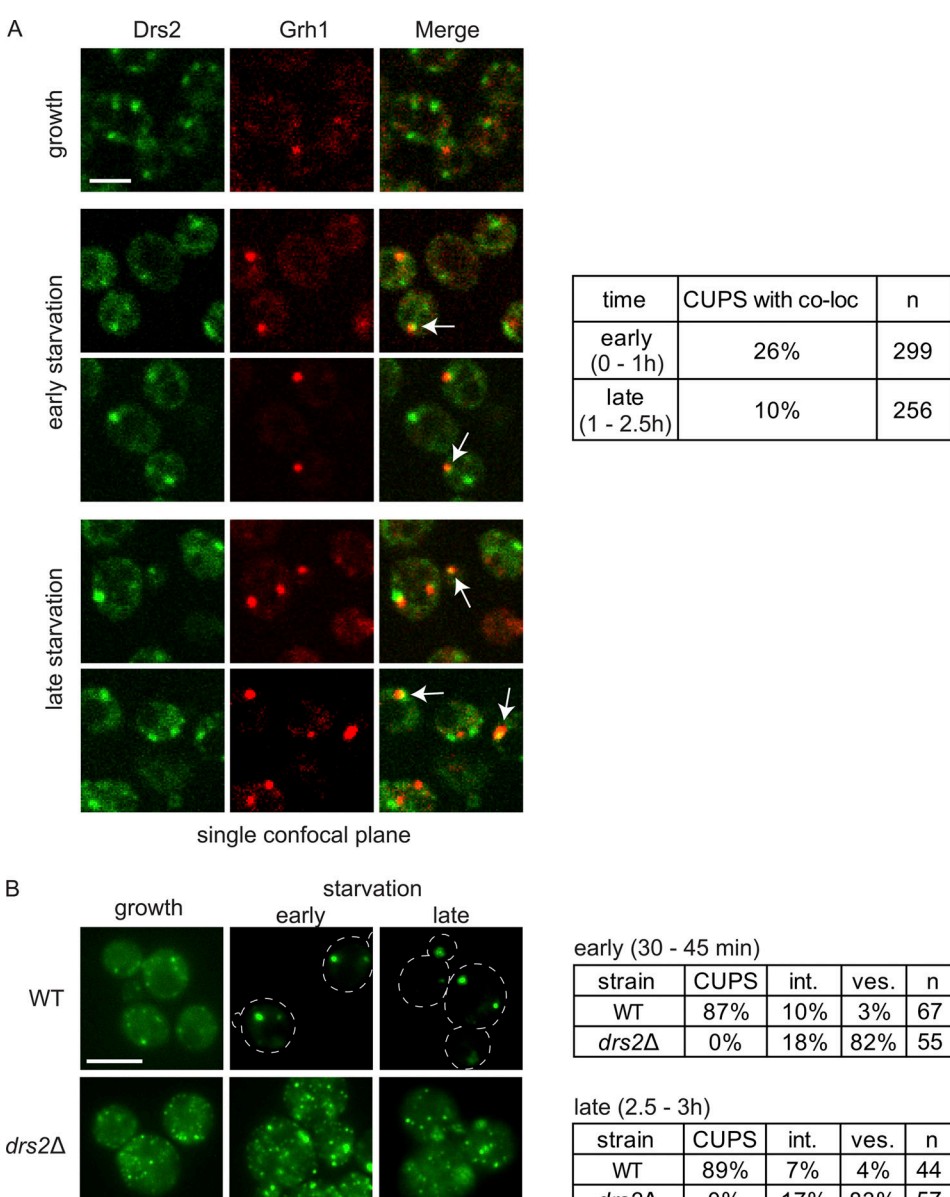

| time | CUPS with co-loc | n |
|---|---|---|
| early (0 - 1h) | 26% | 299 |
| late (1 - 2.5h) | 10% | 256 |

single confocal plane

**early (30 - 45 min)**

| strain | CUPS | int. | ves. | n |
|---|---|---|---|---|
| WT | 87% | 10% | 3% | 67 |
| drs2Δ | 0% | 18% | 82% | 55 |

**late (2.5 - 3h)**

| strain | CUPS | int. | ves. | n |
|---|---|---|---|---|
| WT | 89% | 7% | 4% | 44 |
| drs2Δ | 0% | 17% | 83% | 57 |

Figure 1.   **Drs2 is required for CUPS biogenesis. (A)** Cells genomically expressing Drs2-3xGFP and Grh1-2xmCherry were visualized by confocal spinning-disk microscopy in growth conditions and starvation by incubation in 2% potassium acetate. Short movies were acquired at 10-s intervals to assess the frequency and duration of colocalization. Scale bar = 2 μm. **(B)** WT and *drs2Δ* cells expressing Grh1-2xGFP were visualized by epifluorescence microscopy in growth conditions and after incubation in 2% potassium acetate for the indicated times. Cells were classified with normal CUPS (one to three larger foci per cell); intermediate CUPS ("int."), where a large focus is observed in addition to smaller structures; and vesiculated CUPS ("ves."), where only small foci of Grh1 are observed. Scale bar = 2 μm.

## CUPS formation requires v-SNARE function

A major known function of the Drs2-Rcy1 pathway is the recycling of the exocytic v-SNARE Snc1. We investigated whether Snc1 is also required for CUPS formation. Snc1 and Snc2 are the only post-Golgi v-SNAREs in yeast, forming an essential pair with redundant functions in cell growth and secretion (Protopopov et al., 1993). Deletion of either gene individually produced no phenotype in CUPS formation (Fig. S2). However, a double mutant temperature-sensitive strain lacking Snc1 and expressing a mutant Snc2 (inefficiently recycled from the plasma membrane), snc2-V39A,M42A (Shen et al., 2013), exhibited the highly vesiculated CUPS phenotype, even without a temperature shift (Fig. 2). While the double mutant cells grew under normal conditions, the sole mutated v-SNARE, Snc2, did not support CUPS formation upon starvation.

Is the defect in CUPS formation due to a failure in the recycling of v-SNAREs to the TGN, or is it more directly related to the function of Drs2/Rcy1? Yeast possess a minimal endomembrane system (Day et al., 2018), and the TGN can be classified into an early TGN, which serves as the site for endocytosis and endosomal recycling, marked specifically by the presence of the

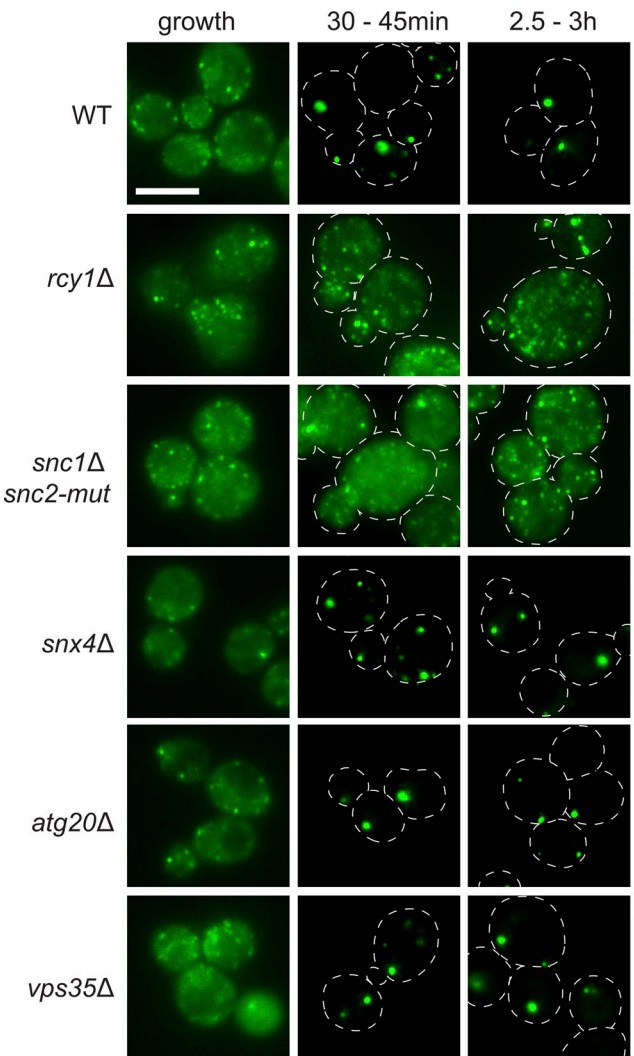

| growth | 30 - 45min | 2.5 - 3h |
| --- | --- | --- |
| WT | | |
| rcy1Δ | | |
| snc1Δ snc2-mut | | |
| snx4Δ | | |
| atg20Δ | | |
| vps35Δ | | |

early (30 - 45min)

| strain | CUPS | int. | ves. | n |
| --- | --- | --- | --- | --- |
| WT | 86% | 7% | 7% | 41 |
| *rcy1 Δ* | 0% | 14% | 86% | 37 |
| *snc1/2-mut* | 0% | 15% | 85% | 34 |
| *snx4 Δ* | 75% | 16% | 9% | 44 |
| *atg20 Δ* | 76% | 11% | 13% | 38 |
| *vps35 Δ* | 70% | 19% | 11% | 53 |

late (2.5 - 3h)

| strain | CUPS | int. | ves. | n |
| --- | --- | --- | --- | --- |
| WT | 86% | 10% | 4% | 61 |
| *rcy1 Δ* | 0% | 10% | 90% | 50 |
| *snc1/2-mut* | 0% | 7% | 93% | 41 |
| *snx4 Δ* | 72% | 21% | 7% | 53 |
| *atg20 Δ* | 74% | 11% | 15% | 35 |
| *vps35 Δ* | 63% | 30% | 7% | 68 |

Figure 2. **Drs2-Rcy1 pathway and the v-SNAREs, Snc1 and Snc2, are required for CUPS formation.** WT and the indicated deletion or mutant strains expressing Grh1-2xGFP were visualized by epifluorescence microscopy in growth conditions and after incubation in 2% potassium acetate for the indicated times. Cells were classified with normal CUPS (one to three larger foci per cell); intermediate CUPS (int.), where a large focus is observed in addition to smaller structures; and vesiculated CUPS (ves.), where only small foci of Grh1 are observed. Scale bar = 2 µm.

t-SNARE Tlg2; and a later TGN, which is responsible for exocytosis and clathrin-coated vesicle formation (Tojima et al., 2019; Toshima et al., 2023). The recycling of Snc1 to the early TGN has been shown to follow three distinct pathways: the Drs2-Rcy1 pathway, which sorts Snc1 from the early endosome-like TGN; the sorting nexins Snx4 and Atg20, along with the retromer, which sort v-SNAREs at late endosomes (or the pre-vacuolar compartment in yeast); and the retromer pathway, which likely becomes significant only when the other two pathways are defective (Best et al., 2020; Hanamatsu et al., 2014; Ma and Burd, 2019). We tested these pathways in CUPS biogenesis and observed no defect in cells lacking the sorting nexins Snx4 and Atg20, or the retromer subunit Vps35 (Fig. 2). Therefore, the defect observed in Drs2/Rcy1-deleted cells is not merely due to the loss of the v-SNARE pool at the TGN. The combined data suggest that CUPS specifically acquire membranes from an early endosome-like TGN compartment in a Drs2/Rcy1-dependent manner.

**Rcy1 and Snc1/Snc2 are required for unconventional secretion**
Unconventionally secreted proteins in yeast can become trapped in the cell wall or periplasmic space. To measure this secretion, a mild cell wall extraction procedure is necessary to prevent cell lysis associated with perturbations to the rigidity of the cell wall (Curwin et al., 2016). Consequently, any genetic mutations or treatments (such as temperature shifts) that exacerbate the problem of lysis cannot be tested using this assay for unconventional secretion. Cells lacking Drs2 exhibit numerous defects, particularly in lipid homeostasis (Hankins et al., 2015), and therefore could not be tested (data not shown). However, *rcy1Δ* cells do not exhibit as many defects associated with the loss of Drs2 function; thus, we tested their capacity to secrete unconventional cargoes such as Acb1 and the antioxidants SOD1 and Trx2.

WT and *rcy1Δ* cells were starved for 2.5 h, after which the secreted material was extracted from the cell wall, as previously described (Curwin et al., 2016). The intracellular and secreted

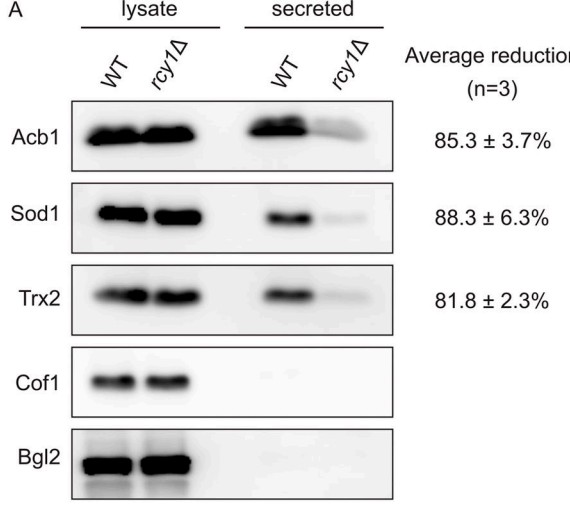

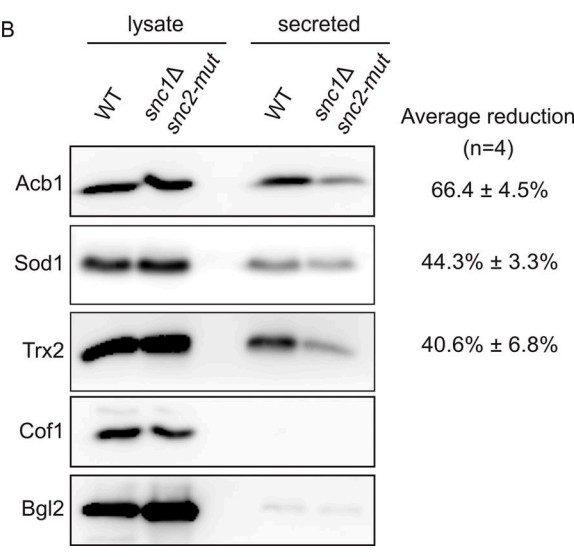

Figure 3. **Rcy1 and v-SNAREs are required for unconventional secretion. (A and B)** WT, *rcy1Δ*, or *snc1Δ snc2-V39A,M42A* cells were grown in the logarithmic phase, washed twice, and cultured in 2% potassium acetate for 2.5 h. The cell wall proteins were extracted from an equal number of cells followed by precipitation with TCA ("secreted"). Lysates and secreted proteins were analyzed by western blot, and the ratio of the secreted to lysate for the indicated protein was determined and compared with that of WT in each experiment. Statistical analyses were performed for the indicated unconventional cargo proteins, and the reduction in secretion compared with WT is indicated as ± SD. Source data are available for this figure: SourceData F3.

fractions were analyzed by western blot for various cargoes, including Cof1, which is used to monitor cell lysis, and the known cell wall protein Bgl2, which measures traffic to the cell surface along the conventional secretory pathway (Curwin et al., 2016). Loss of Rcy1 resulted in a strong defect in the release of Acb1, SOD1, and Trx2, without causing the release of the cytoplasmic content, as indicated by the absence of Cof1 (Fig. 3 A). Similarly, the Snc1 and Snc2 double mutant, which is defective in CUPS formation (Fig. 2), was tested and also exhibited a reduction in the secretion of Acb1, SOD1, and Trx2 compared with control cells (Fig. 3 B). Therefore, we conclude that

unconventional secretion during starvation requires Rcy1 (presumably in concert with Drs2) and v-SNARE activity.

## Drs2, Snc2, and Tlg2 are contained in a modified TGN that transiently contacts CUPS

We generated N-terminal GFP fusions of the v-SNAREs Snc1 and Snc2, along with their cognate t-SNARE Tlg2, which preferentially labels the early TGN and likely receives the v-SNARE vesicles being recycled in a Drs2-Rcy1–dependent manner. Most analyses of v-SNARE location and trafficking have been conducted using highly overexpressed N-terminal GFP-tagged Snc1, which, at the steady state, primarily labels the plasma membrane of growing buds and some internal structures (Lewis et al., 2000). However, Graham and colleagues developed an mNG-Snc1 construct that can be visualized at much lower expression levels, which preferentially labeled the TGN and endosomes, indicating that the overexpression of these SNAREs leads to aberrant localization at steady-state conditions (Best et al., 2020). To avoid plasmid overexpression altogether, we integrated the GFP tag at the N terminus of each SNARE at its endogenous locus under the control of the Sed5 promoter.

Interestingly, when the v-SNAREs were expressed at this lower more endogenous-like level, we observed distinct steady-state localization patterns for Snc1 and Snc2 during growth. The overall signal of Snc1 was weaker than that of Snc2, with distinct localization of Snc1 seen only at the tips of very small-budded cells and partially in the neck of larger budded cells (Fig. 4 A). In unbudded and large/medium-budded cells, Snc1 displayed mostly a diffuse signal. In contrast, Snc2 exhibited clear localization in all cells, preferentially labeling internal structures, the neck of large-budded cells, and occasionally the plasma membrane of small- and medium-budded cells (Fig. 4 A). Although these v-SNAREs are redundant in some functions, they exhibit distinct steady-state itineraries. Tlg2 localized, as expected, to four to six punctate structures per cell during growth (Fig. 4 A). Notably, none of these SNARE proteins could be colocalized with Grh1 under growth conditions.

Upon starvation, the Snc1 signal rapidly became diffuse in most cells (<5% retained one to two faint foci). In contrast, the Snc2 signal remained high in all cells, labeling fewer but larger punctate elements (Fig. 4 B). Tlg2, similar to Drs2, also labeled fewer and larger structures immediately upon starvation. Both Snc2 and Tlg2 structures transiently colocalized with Grh1 (Fig. 4 B). In the case of Snc2, this colocalization was observed in an average of 13% of cells at any time point during starvation, while Tlg2 colocalization with Grh1 was more frequent early in starvation (14% of cells early and 7% later), similar to Drs2. Examination of GFP-Tlg2 with Drs2-3xCherry revealed that they indeed reside in the same compartment during starvation (Fig. S3). Drs2 is predicted to be present in both early and late TGN membranes due to its roles in anterograde and retrograde transport, while Tlg2 is specifically located in the early TGN. We observed partial colocalization of Drs2 and Tlg2 under growth conditions (Fig. S3). During starvation, the number of TGN membranes was reduced, leading to significantly increased colocalization of Drs2 and Tlg2 (Fig. S3). A similar pattern was observed when mCherry-Snc2 and GFP-Tlg2 were examined

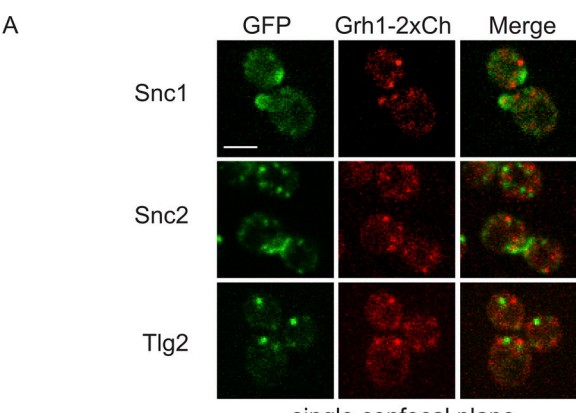

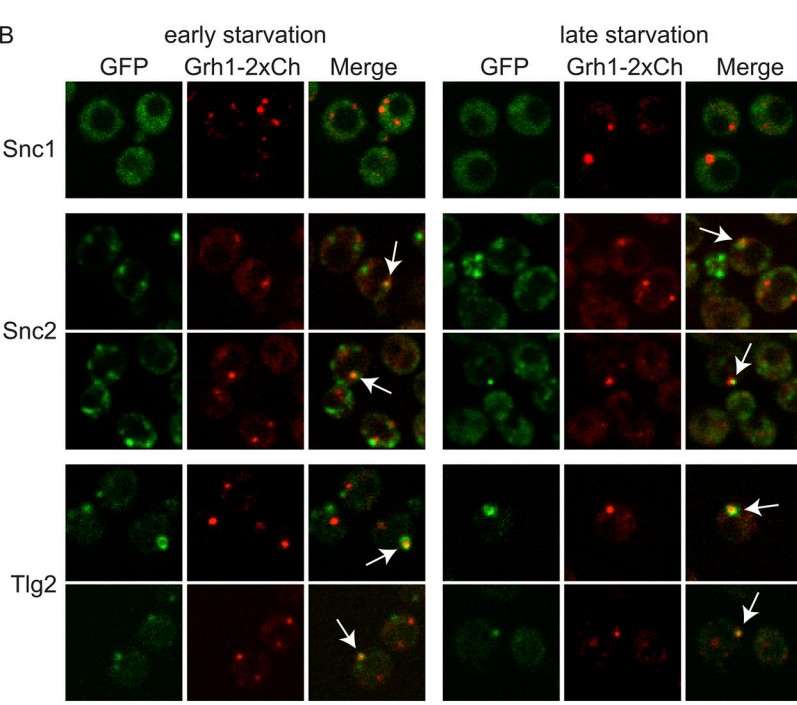

Figure 4. **CUPS contain a pool of the v-SNARE Snc2 and the t-SNARE Tlg2. (A and B)** Cells genomically expressing Grh1-2xCherry with GFP-Snc1, GFP-Snc2, or GFP-Tlg2 were visualized by confocal spinning-disk microscopy in growth conditions and (B) throughout the time course of culture in 2% potassium acetate. Short movies were acquired at 10-s intervals to assess the frequency and duration of colocalization. Scale bar = 2 µm.

| | Snc2 | | | | Tlg2 | |
|---|---|---|---|---|---|---|
| time | CUPS with co-loc | n | | time | CUPS with co-loc | n |
| early (0 - 1h) | 14% | 125 | | early (0 - 1h) | 14% | 110 |
| late (1 - 2.5h) | 12% | 139 | | late (1 - 2.5h) | 7% | 136 |

during starvation. Although the signal from mCherry-Snc2 was weak compared with the GFP version, larger Snc2 structures were observed to colocalize with Tlg2 (Fig. S3).

## SCLIM reveals the highly dynamic events leading to CUPS formation

We previously characterized the ultrastructure of CUPS using correlative light–electron microscopy as a spherical tubulovesicular structure that increases in size during starvation and acquires large, enveloping cup-shaped cisternae (Curwin et al., 2016). To gain deeper insight into the organization of the membranes that comprise CUPS and their potential interactions

with the modified TGN, we used SCLIM (Kurokawa et al., 2013, 2019). SCLIM analysis of Grh1-2xGFP confirmed the presence of these structures and further revealed their dynamic behavior (Fig. 5; and Videos 1, 2, 3, and 4). Grh1 was localized to numerous small, mobile structures under growth conditions. Detailed examination of larger Grh1-positive structures after 3 h of starvation revealed mature CUPS that could be categorized into three forms: spherical, cuplike, and curved (Fig. 5, B and C). Although the overall mobility of these structures decreased over the course of starvation, the morphology of the CUPS dynamically changed among the different forms (Fig. 5 D and Video 1). Dynamic Grh1 structures were observed to occasionally contact one another,

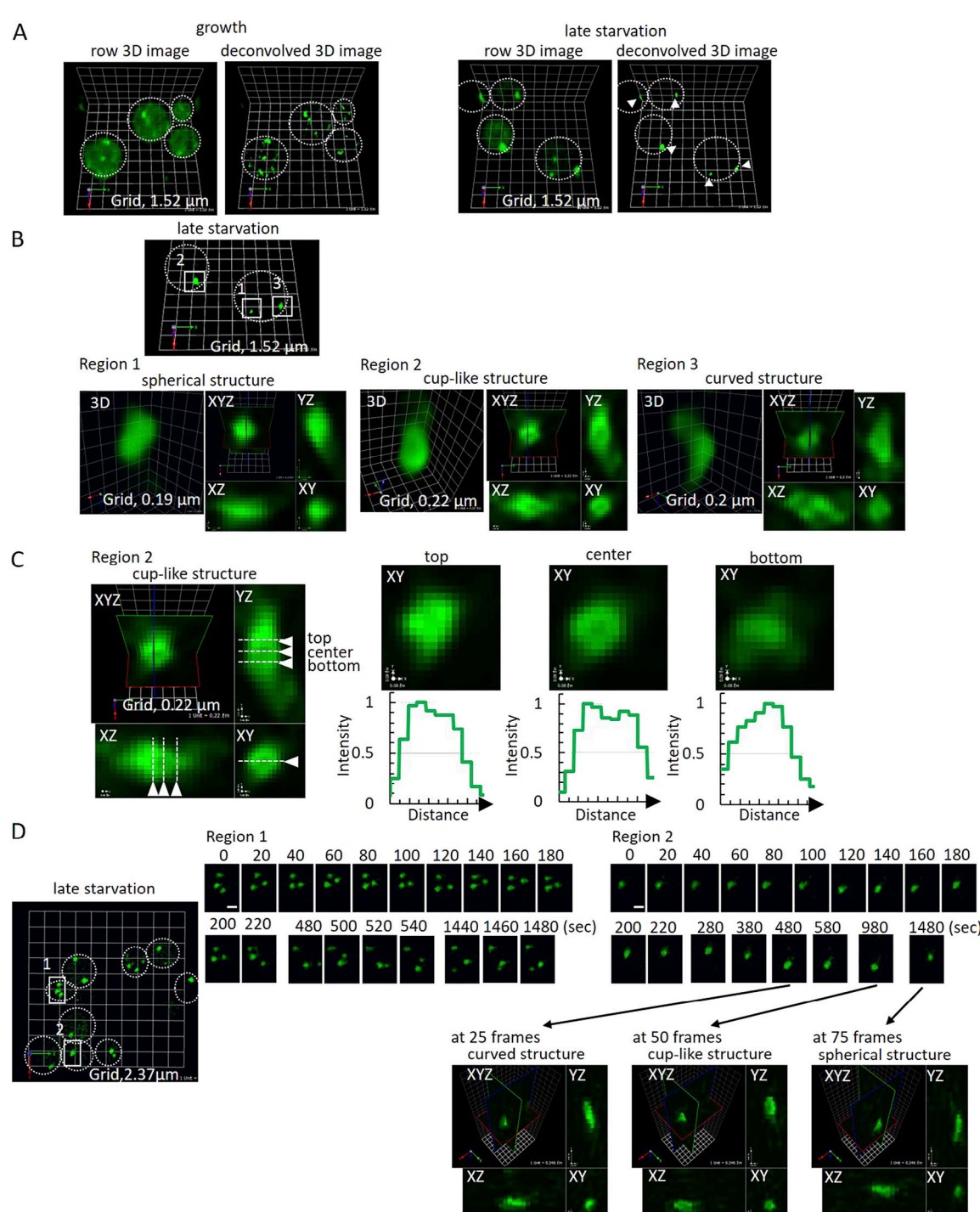

Figure 5. **SCLIM reveals the dynamic structure of CUPS. (A)** Grh1-2xGFP cells were incubated in normal growth conditions or 2% potassium acetate for 3 h and visualized by SCLIM. In growth, Grh1 labeled many small and mobile structures (early Golgi membranes and ER exit sites). In starvation, Grh1 labeled fewer, larger, and less mobile membrane structures (CUPS) (arrowheads). Grid = 1.52 μm. **(B and C)** Line-scan analysis in 3D of multiple CUPS structures revealed three forms: spherical (3/14), complex curved (8/14), or cup-shaped (3/14). **(D)** Visualization of CUPS over time showed stable, mature CUPS are still dynamic, able to change morphology between the different forms. Region 1 = moving structures; Region 2 = nonmoving structure. The time-lapse images of Region 2 at 25, 50, and 75 frames are shown in XYZ images. Scale bar = 1 μm.

possibly fusing and becoming more stable (Fig. 5 D; and Videos 1, 2, 3, and 4). Moreover, analysis conducted earlier in the starvation period (1–1.5 h) indicated that dynamic Grh1-positive structures frequently contacted each other, likely growing in size through fusion, while larger Grh1 structures were seen to fragment at

times (Videos 5, 6, and 7). Overall, the SCLIM analyses reveal that CUPS form through dynamic interactions among Grh1-containing membranes, likely involving both fusion and fission. These highly dynamic interactions ultimately reach a steady state, generating a more stable CUPS (Videos 1, 2, 3, and 4).

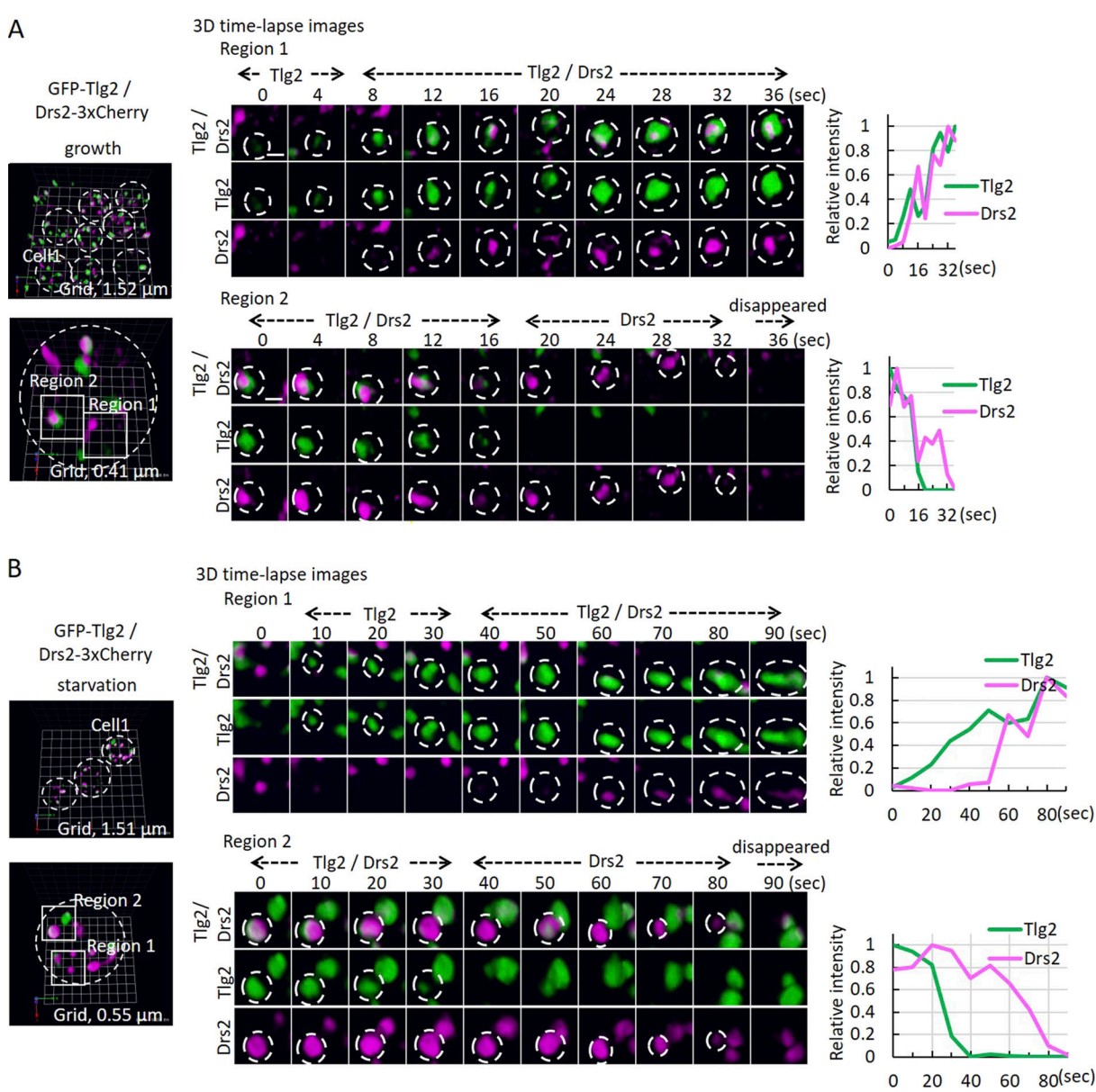

Figure 6. **SCLIM analysis of Drs2 and Tlg2 labeled structures in growth (TGN) and starvation (modified TGN). (A)** Drs2-3xCherry (magenta) and GFP-Tlg2 (green) cells were visualized in the growth condition. Time-lapse images of the two regions are indicated. Scale bar = 0.5 μm. Relative fluorescence intensities of Drs2-3xCherry and GFP-Tlg2 in the cisterna are shown on the right. **(B)** Drs2-3xCherry (magenta) and GFP-Tlg2 (green) cells were visualized at 2-h starvation. Time-lapse images of the two regions are indicated. Scale bar = 0.5 μm. Relative fluorescence intensities of Drs2-3xCherry and GFP-Tlg2 in the cisterna are shown on the right.

### Dynamics of the modified TGN and its contacts with CUPS

We next examined the organization of Drs2-3xCherry and GFP-Tlg2 during growth and starvation using SCLIM. Tlg2 localizes to an early TGN compartment under growth conditions. Colocalization of Drs2 and Tlg2 was observed in some Golgi cisternae, but not all. Tracking these structures over time revealed a sequence of events: structures were first labeled by Tlg2, followed by the acquisition of Drs2 (colocalization), and eventually, Tlg2 was lost first, followed by Drs2 (Fig. 6 A and Video 8).

During starvation, the extent of colocalization between Drs2 and Tlg2 significantly increased in the modified TGN. Interestingly, we again observed Tlg2 appearing first, followed by Drs2. Subsequently, Tlg2 was lost, followed by Drs2, until eventually

the modified TGN was no longer detectable (Fig. 6 B; and Videos 9 and 10).

### The modified TGN is ultimately fragmented through its contact with CUPS

We analyzed the dynamics of Grh1-2xmCherry-containing CUPS in combination with Drs2-3xGFP, GFP-Tlg2, or GFP-Snc2 to visualize the interactions between CUPS and the modified TGN. SCLIM analysis captured numerous contacts between these structures. Although these contacts were infrequent, we do not expect to capture all transient interactions. The nature of these contacts often involved the insertion of a modified TGN tubule into CUPS, observable with Drs2, Tlg2, or Snc2 as the modified

TGN markers. In Fig. 7 A and Video 11, we provide an example where Drs2-positive membranes insert into CUPS, resulting in the production of fragments from Drs2-positive elements. In another instance, using Tlg2 as the modified TGN marker, the CUPS collar appears to sever a tubule derived from the modified TGN, although we cannot conclusively state that CUPS directly cut the modified TGN based on this analysis alone (Fig. 7 B and Video 12). In addition to the modified TGN, Snc2—and to a lesser extent Drs2—also labeled numerous smaller structures that frequently contacted or were near CUPS (Fig. S4; and Videos 13, 14, 15, 16, and 17). These are likely vesicles or tubules, as Drs2 itself has been shown to be packaged into vesicles during its activity (Liu et al., 2008). The combined evidence suggests that the modified TGN is consumed during starvation in the process of carrier formation, driven at least partially by the action of Drs2-Rcy1. While these carriers do not appear to fuse directly with CUPS—as indicated by our SCLIM observations—the absence of Drs2-Rcy1 activity is essential for both CUPS formation and unconventional secretion.

## Discussion

George Palade mapped the pathway of protein secretion, establishing a foundation for decades of research on how proteins are exported from the ER and transported through the Golgi complex to their ultimate destinations. Gunter Blobel elucidated how proteins enter this ER–Golgi pathway via the N-terminal signal sequence. Recently, attention has begun to turn toward the understanding that eukaryotic cells can secrete proteins that do not follow the Blobel–Palade pathway. A diverse class of proteins, essential for various physiological functions in the extracellular space—such as immune surveillance, tissue reorganization, insulin homeostasis, and protection from oxidative damage—are released through unconventional means.

We demonstrated that the Golgi-associated protein GrpA (Grh1 in yeast; GORASP—present as a single gene in invertebrates and two in vertebrates) is necessary for the secretion of Acb1 in nutrient-starved *Dictyostelium discoideum* (Kinseth et al., 2007). Since then, this function of GrpA has been shown to be conserved. Our research has focused on the Grh1-dependent unconventional secretion pathway, leading to the identification of a compartment we term CUPS, which forms under conditions that trigger unconventional secretion (Bruns et al., 2011; Cruz-Garcia et al., 2014). CUPS, characterized by the presence of Grh1, require PI4P for their biogenesis (Curwin et al., 2016). Here, we have demonstrated that the significance of PI4P lies in the necessity of its effector, Drs2—a transmembrane aminophospholipid flippase—for both CUPS formation and unconventional secretion. Among the many proteins that collaborate with Drs2 in trafficking at the TGN, only Rcy1 is implicated in unconventional secretion. This pathway, under growth conditions, functions to recycle the v-SNARE Snc1. Our data reveal that the v-SNARE functions of the Snc1 and Snc2 orthologous pair are also essential for CUPS formation and unconventional secretion. Additionally, our findings indicate that starving yeast generates a modified form of the TGN that includes Drs2, the v-SNARE Snc2 (but not Snc1), and the t-SNARE Tlg2. These findings underscore the necessity of membranes derived from both the early and late Golgi complex, as well as the proteins required for membrane fusion, in the secretion of signal sequence–lacking cytoplasmic proteins that cannot enter the ER.

### CUPS contact the modified TGN during unconventional protein secretion

CUPS often appear to enwrap or encircle the modified TGN. We captured a fascinating event following this contact: a tubule emerging from the modified TGN is collared by CUPS and appears to be severed. This event is reminiscent of the interactions between the ER and endosomes, specifically in the fission of the latter compartment (Rowland et al., 2014). Additionally, the interaction between CUPS and the modified TGN during starvation resembles the contact between the ERGIC and TGN/recycling endosomes in the Golgi bypass trafficking associated with the egress of coronaviruses in mammalian cells (Saraste et al., 2022; Saraste and Prydz, 2021). Grh1 is located in the cis-most compartment of the Golgi, which can be regarded as the yeast equivalent of the ERGIC under growth conditions (Tojima et al., 2024). Generally, membrane contacts between cellular compartments facilitate the exchange of lipids and ions. A similar exchange of essential molecules is likely necessary for the combined function of CUPS and the modified TGN in unconventional secretion.

### The pathway of unconventional secretion

The proteins secreted by this pathway lack a signal sequence and do not undergo N-linked glycosylation, meaning they cannot—and do not need to—enter the ER. A key issue with this class of cytoplasmic proteins is that they are released in a single bolus, with only a small fraction of the total cellular pool being exported. Consequently, cells must possess a mechanism to selectively collect a portion of their secretory clients from the total pool.

The signals involved in this sorting process are not fully understood; however, for proteins such as Acb1, a diacidic motif is essential for its secretion (Cruz-Garcia et al., 2020). Acb1 has been localized to CUPS through immunoelectron microscopy in yeast, as well as in a protease-protected membrane-bounded compartment in *Dictyostelium discoideum* (Cabral et al., 2010; Curwin et al., 2016). We propose that CUPS and the modified TGN function collaboratively to collect and sort proteins such as Acb1. Notably, it has recently been demonstrated that interferon-dependent secretion of mature IL-1β involves the translocation of this cytokine into the TGN en route to secretion in monocytes (Caielli et al., 2024). The trafficking of Acb1 is shown to depend on the endosomal sorting complex required for transport (ESCRT), although Vps4 is not necessary. Snf7, an ESCRT-III protein, transiently localizes to the CUPS-modified TGN interface and may play a role in generating vesicles or tubules enriched in Acb1 for trafficking to the cell surface. The fusion of such an Acb1-containing intermediate with the plasma membrane would facilitate the delivery of cargo to the extracellular space. The remaining modified TGN fragments persist. Upon returning to nutrient-rich conditions, the modified TGN

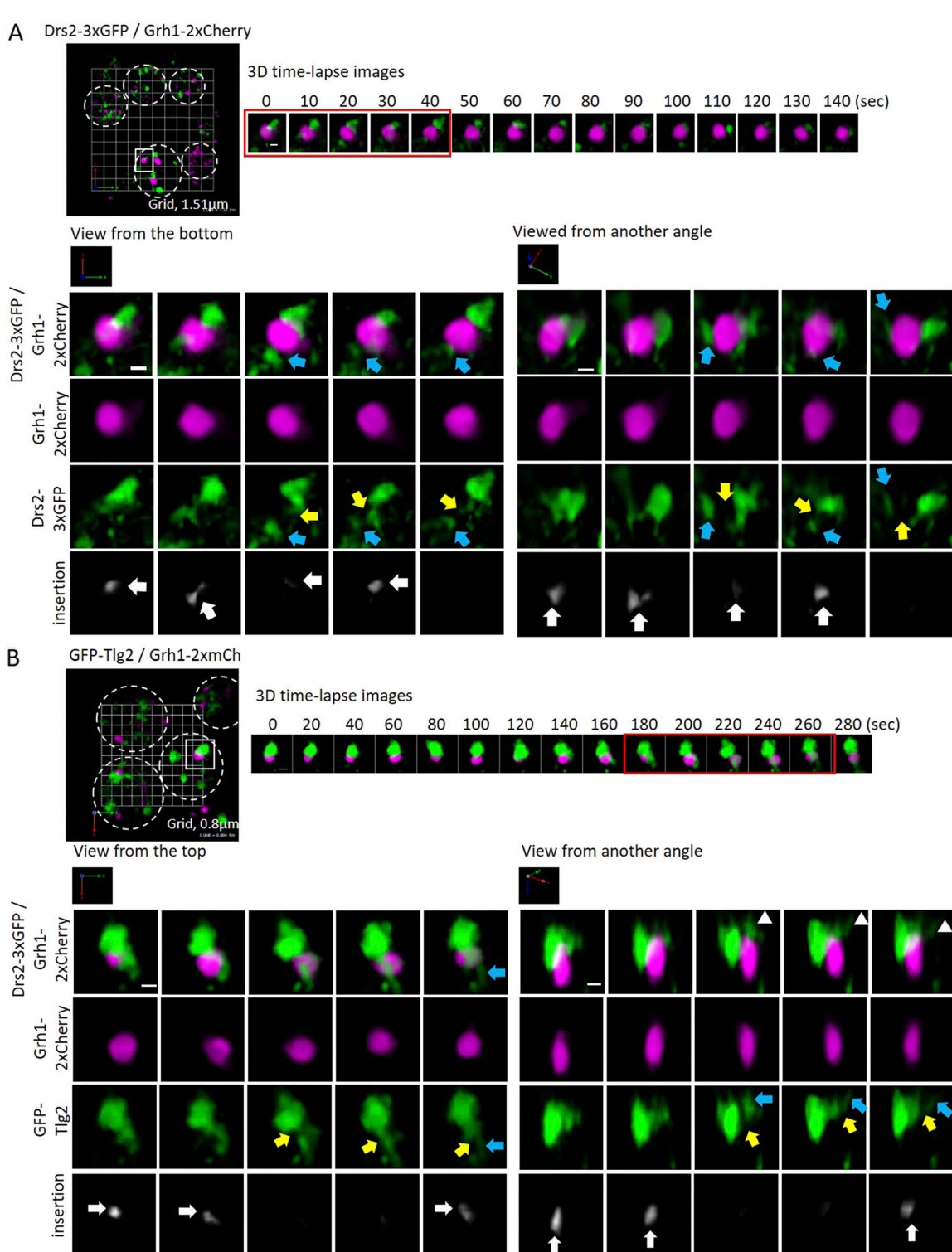

Figure 7. **SCLIM analysis of CUPS-modified TGN contacts. (A)** Grh1-2xmCherry (magenta) and GFP-Tlg2 (green) cells cultured in the starvation condition for 1.5 h. 3D time-lapse images (10-s intervals) are indicated. Light blue arrows show separated membrane structures labeled with GFP-Tlg2. Yellow arrows indicate where the membrane structures have been cut. White arrows indicate where Grh1 contacts with a Tlg2 protrusive membrane. Scale bar = 0.5 μm. **(B)** Grh1-2xCherry (magenta) and Drs2-3xGFP (green) cells cultured in the starvation condition for 1.5 h. 3D time-lapse images (20-s intervals) are indicated. Light blue arrows show separated membrane structures labeled with Drs2-3xGFP. Yellow arrows indicate where the membrane structures have been cut. White arrows indicate where Grh1 contacts with a Drs2 protrusive membrane. Scale bar = 0.5 μm.

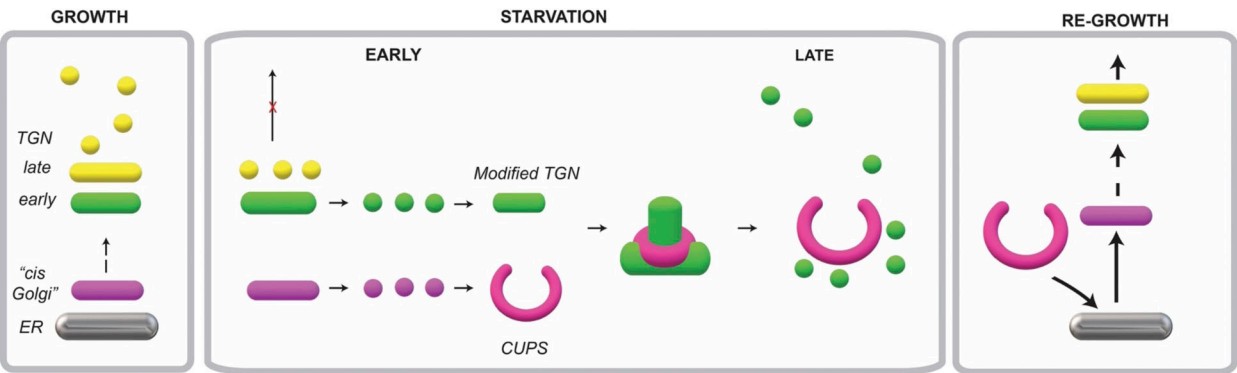

**Figure 8. Working scheme building CUPS-modified TGN for unconventional secretion.** During growth, cells predominantly depend on the conventional ER–Golgi pathway of protein secretion. When cells are cultured in starvation medium, there is a sharp reduction in the use of conventional secretory pathway and the cells switch to a new or an unconventional mode to release essential proteins to the cell's exterior. A cis-Golgi membrane produces small fragments, which do not contain glycosylation enzymes, in a COPI-independent manner to synthesize CUPS (magenta). The early TGN produces small membranes to generate a compartment that we have called the modified TGN (green). Our data show that tubules emanating from the modified TGN are collared by CUPS, which is followed by severing of the tubule. We suggest that these contacts, over a period, lead to the consumption of the modified TGN to produce smaller elements (vesicles + tubules). These smaller elements are likely used for delivering essential proteins to other compartments of the cell and releasing proteins such as SOD1 and Acb1 to the cell's exterior. This mode of TGN consumption is common to both the conventional and unconventional protein secretion processes. Upon shifting cells to growing conditions, components of the CUPS are delivered by COPI vesicles to the ER, which then traffic the respective components to the Golgi, thereby restoring the Golgi to restart the conventional mode of protein secretion.

remnants and CUPS fuse with the ER, allowing the conventional secretory pathway to resume and rebuild a standard Golgi apparatus (Fig. 8).

## Materials and methods

### Yeast strains and media

Yeast cells were grown in synthetic complete (SC) media (0.67% yeast nitrogen base without amino acids, 2% glucose/galactose supplemented with amino acid drop-out mix from ForMedium). All strains are derived from the BY4741 background (*MATa his3Δ1 leu2Δ0 met15Δ0 ura3Δ0*). Deletion strains were from the EUROSCARF collection with individual genes replaced by KanMx4. Strains expressing C-terminally 2xyeGFP- and/or 2xyomCherry-tagged Grh1 were constructed by a PCR-based targeted homologous recombination and have been described previously (Cruz-Garcia et al., 2014). In many cases, strains were generated by mating and sporulation, followed by selection of clones with appropriate markers, and confirmation of haploidy. The double mutant v-SNARE (BY4741 *snc1Δ::kanMX4 snc2-V39A,M42A*) strain was provided by Peter Novick (Department of Cellular and Molecular Medicine, University of California San Diego, La Jolla, CA, USA). This mutant expressing Grh1-2xyeGFP was generated by mating, sporulation, and confirmation of markers and temperature sensitivity. Drs2-3xGFP, KanMx4 strain was a gift from Oriol Gallego (UPF, Barcelona, Spain). To generate the strain used for the flippase mutant experiments, we changed the endogenous Drs2 promoter in the Drs2-3xGFP, Grh1-2xyomCherry strain to the glucose-repressible pGalS promoter using the pYM-N31 toolbox plasmid (Janke et al., 2004). Positive clones were propagated and further transformed in a galactose-containing medium with plasmids either expressing no protein (empty vector, pRS415), WT Drs2 (pRS315-Drs-WT, Chen et al., 1999), or mutant Drs2 (pRS315-Drs2-D560N, Chen et al., 1999).

### Construction of N-terminally tagged SNAREs

The plasmid pYM-N9 (PCR toolbox) was used to generate a new template vector for PCR-based integration containing the NatNT2 selection cassette, the promoter of Sed5, followed by two tandem yeGFP. The promoter of Sed5 was amplified from genomic DNA with primers "SacI PrSed5 Fw1": 5′-ATAGAGCTC TTACCATGTCCTCCAGAATTACGA-3′ and "XbaI PrSed5 Rv1": 5′-TCATCTAGAGGGAGTTGTGTGGTATGGTG-3′ to generate a 658-bp fragment and was cloned into pYM-N9, replacing the high-expression *ADH1* promoter. Subsequently, a second yeGFP fragment was generated using primers "XbaI ATG yeGFP": 5′-TGATCTAGAAAAAATGTCTAAAGGTGAAGAATTATTCACTGG-3′ and "EcoRV non-stop yeGFP": 5′-TCTGATATCAGGCCTCAT CGATGAATTCTCTGTCGGA-3′ and cloned downstream of the first yeGFP. Finally, standard S1/S4 primers were used to generate the N-terminal integration fragment targeting the Snc1, Snc2, and Tlg2 loci. Strains were confirmed as positive by microscopy and PCR to confirm the presence of 2 yeGFP. In most cases, however, only one GFP integrated and the resulting 1xyeGFP strains were used. In the case of Snc1, 2xyeGFP was initially analyzed, and subsequently, a single yeGFP version was generated and found to behave in an identical manner as the 2xyeGFP version. SnapGene software (available at https://www.snapgene.com; GSL Biotech) was used for molecular cloning design.

### Antibodies

All antibodies were raised in rabbit and have been described previously. Anti-SOD1 and anti-Trx2 were the kinds of gifts of Yoshiharu Inoue (Research Institute for Food Science, Kyoto University, Kyoto, Japan) and T. O'Halloran (Northwestern University, Chicago, IL, USA), respectively. Anti-Cof1 was kindly provided by John Cooper (Washington University in St. Louis, St. Louis, MO, USA), and anti-Bgl2 was a gift from Randy

Schekman (UC Berkeley, Berkeley, CA, USA). Anti-Acb1 antibody was generated by inoculating rabbits with recombinant, untagged Acb1, was purified from bacteria, and has been described previously (Curwin et al., 2016). HRP-conjugated anti-rabbit secondary was from Jackson ImmunoResearch (Cat# 711-035-152).

## Cell wall extraction assay
Yeast cells were inoculated at a density of 0.003–0.006 $OD_{600}$/ml in SC medium at 25°C. The following day, when cells had reached $OD_{600}$ of 0.4–0.7, equal numbers of cells (16 $OD_{600}$ U) were harvested, washed twice in sterile water, resuspended in 1.6 ml of 2% potassium acetate, and incubated for 2.5 h. When growing cells were to be analyzed, 16 $OD_{600}$ U were directly harvested. The cell wall extraction buffer (100 mM Tris-HCl, pH 9.4, 2% sorbitol) was always prepared fresh before use and kept on ice. To ensure no loss of cells and to avoid cell contamination in the extracted buffer, 2-ml tubes were siliconized (Sigmacote) prior to collection. Cells were harvested by centrifugation at 3,000 × $g$ for 3 min at 4°C, medium or potassium acetate was removed, and 1.6 ml of cold extraction buffer was added. Cells were resuspended gently by inversion and incubated on ice for 10 min, after which they were centrifuged as before, 3,000 × $g$ for 3 min at 4°C, and 1.3 ml of extraction buffer was removed to ensure no cell contamination. The remaining buffer was removed, and the cells were resuspended in 0.8 ml of cold TE buffer (Tris-HCl, pH 7.5, EDTA) with protease inhibitors (aprotinin, pepstatin, leupeptin [Sigma-Aldrich]), and 10 μl was boiled directly in 90 μl of 2x sample buffer (lysate). For western blotting analysis, 30 μg of BSA (Sigma-Aldrich) carrier protein and 0.2 ml of 100% TCA (Sigma-Aldrich) were added to the extracted protein fraction. Proteins were precipitated on ice for 1 h, centrifuged at 16,000 × $g$ for 30 min, and boiled in 50 μl 2x sample buffer. For detection, proteins (10 μl each of lysate or wall fractions) were separated in a 12% polyacrylamide gel before transfer to 0.2-μm nitrocellulose membrane (GE Healthcare) for detection by western blotting.

## Epifluorescence microscopy
After incubation in the appropriate medium, cells were harvested by centrifugation at 3,000 × $g$ for 3 min, resuspended in a small volume of the corresponding medium, spotted on a microscopy slide, and imaged live with a DMI6000 B microscope (Leica) equipped with a DFC 360FX camera (Leica) using an HCX Plan Apochromat 100×1.4 NA objective. Images were acquired using LAS AF software (Leica), and processing was performed with ImageJ 1.47n software.

## Spinning-disk confocal fluorescence microscopy
After incubation in starvation medium for 20 min, ∼0.05 $OD_{600}$ nm of cells was plated in starvation medium on concanavalin A–coated (Sigma-Aldrich) Lab-Tek chambers (Thermo Fisher Scientific) and was allowed to settle for 20 min at 25°C. Cells were continuously imaged for up to 10 min throughout starvation. Whole-cell Z-stacks with a step size of 0.4 μm were continuously acquired (10 s frames) using a spinning-disk confocal microscope (Revolution XD; Andor Technology) with a Plan Apochromat 100× 1.45 NA objective lens equipped with a dual-mode electron-modifying charge-coupled device camera (iXon 897 E; Andor Technology) and controlled by iQ Live Cell Imaging software (Andor Technology). Some later images were taken on a newer spinning-disk system (Andor Dragonfly) equipped with a 488-nm and/or 561-nm diode, using a U Plan Apo 60× 1.4 oil objective and an iXon-EMCCD Du-897 camera. A 2× camera zoom was used to reach the Nyquist sampling, and Fusion software was used for acquisition. Postacquisition processing was performed with ImageJ 1.47n software.

## Super-resolution confocal live imaging microscopy (SCLIM)
For high-speed live imaging, yeast cells were immobilized on glass slides using concanavalin A and imaged by SCLIM at room temperature. SCLIM was developed by combining Olympus model IX-71 inverted fluorescence microscope with a UPlanSApo 100× NA 1.4 oil objective lens (Olympus), a high-speed and high signal-to-noise ratio spinning-disk confocal scanner (Yokogawa Electric), a custom-made spectroscopic unit, image intensifiers (Hamamatsu Photonics) equipped with a custom-made cooling system, magnification lens system for giving 266.7× final magnification, and EMCCD cameras (ImagEM; Hamamatsu Photonics) (Kurokawa et al., 2013). For excitation of GFP and Cherry, solid-state lasers emitting at 473 nm (Blues, 50 mW; Cobolt) and 561 nm (Jive, 50 mW; Cobolt) were used, respectively. Image acquisition was executed by custom-made software (Yokogawa Electric). For 3D images, we collected optical sections spaced 100 nm apart in stacks by oscillating the objective lens vertically with a custom-made piezo actuator. Z-stack images were converted to 3D voxel data and processed by deconvolution with Volocity software (PerkinElmer) using the theoretical point-spread function for spinning-disk confocal microscopy. Imaging analysis was done using Volocity and MetaMorph software (Molecular Devices).

## Online supplemental material
Fig. S1 shows CUPS biogenesis requires Drs2 flippase activity. Fig. S2 shows CUPS formation requires the Drs2-Rcy1 signaling pathway and the v-SNAREs Snc1 and Snc2. Fig. S3 shows Drs2, Tlg2, and Snc2 label the same compartment in starvation—modified TGN. Fig. S4 shows Drs2 and Snc2 also label small vesicles that contact with or are near CUPS. Videos 1, 2, 3, and 4 show SCLIM analysis of Grh1-2xGFP under growth and starvation conditions. Videos 5, 6, and 7 show CUPS form by dynamic fusion and fission of existing membranes. Video 8 shows colocalization and dynamics of Drs2 and Tlg2 under growth conditions. Videos 9 and 10 show colocalization and dynamics of Drs2 and Tlg2 under starvation conditions. Video 11 shows an example where Drs2-positive membranes insert into CUPS. Video 12 shows SCLIM analysis of CUPS-modified TGN contacts using Tlg2 as a marker. Videos 13, 14, 15, 16, and 17 show Drs2 and Snc2 also label small vesicles that contact with or are near CUPS.

## Data availability
The data underlying this study are available in the published article and its online supplemental material.

## Acknowledgments

We thank members of the Malhotra laboratory for their valuable discussions. V. Malhotra is an Institució Catalana de Recerca i Estudis Avançats professor at the Centre for Genomic Regulation.

This work was funded by grants from the Spanish Ministry of Economy and Competitiveness (BFU2013-44188-P and BFU2016_75372-P to V. Malhotra) and by Grants-in-Aid for Scientific Research from Japan Society for the Promotion of Science (JP17H06420 and JP18H05275 to A. Nakano and K. Kurokawa). We acknowledge support of the Spanish Ministry of Economy, Industry and Competitiveness to the European Molecular Biology Laboratory partnership, the Programmes "Centro de Excelencia Severo Ochoa 2013-2017" (SEV-2012-0208 & SEV-2013-0347), and the Centres de Recerca de Catalunya Programme/Generalitat de Catalunya. This work reflects only the authors' views, and the EU Community is not liable for any use that may be made of the information contained therein. Open Access funding provided by the Universitat Pompeu Fabra.

Author contributions: A.J. Curwin: conceptualization, data curation, formal analysis, investigation, methodology, project administration, validation, visualization, writing—original draft, and writing—review and editing. G. Bigliani: visualization and writing—review and editing. N. Brouwers: investigation, resources, validation, visualization, and writing—original draft. A. Nakano: investigation, supervision, and writing—review and editing. V. Malhotra: conceptualization, data curation, formal analysis, funding acquisition, project administration, resources, supervision, validation, writing—original draft, and writing—review and editing.

Disclosures: The authors declare no competing interests exist.

Submitted: 22 December 2023

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

# Supplemental material

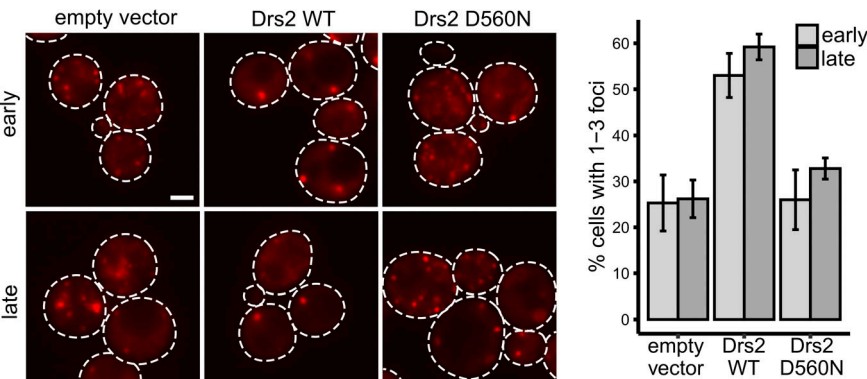

Figure S1.   **Drs2 flippase activity is required for CUPS biogenesis.** Cells genomically expressing pGalS-Drs2-3xGFP and Grh1-2xmCherry were transformed with an empty vector, WT Drs2, or mutant Drs2 (D560N) plasmids in a galactose-containing medium. Cells were switched to glucose medium, in order to repress the expression of Drs2-3xGFP, 72 h prior to starvation in 2% potassium acetate. Cells were visualized by epifluorescence microscopy at the indicated times (early, 30- to 35-min incubation; late, 2-h incubation). **(A)** Representative images of Grh1-positive foci. Scale bar = 2 μm. **(B)** Quantification of the percentage of cells showing one to three foci per cell.

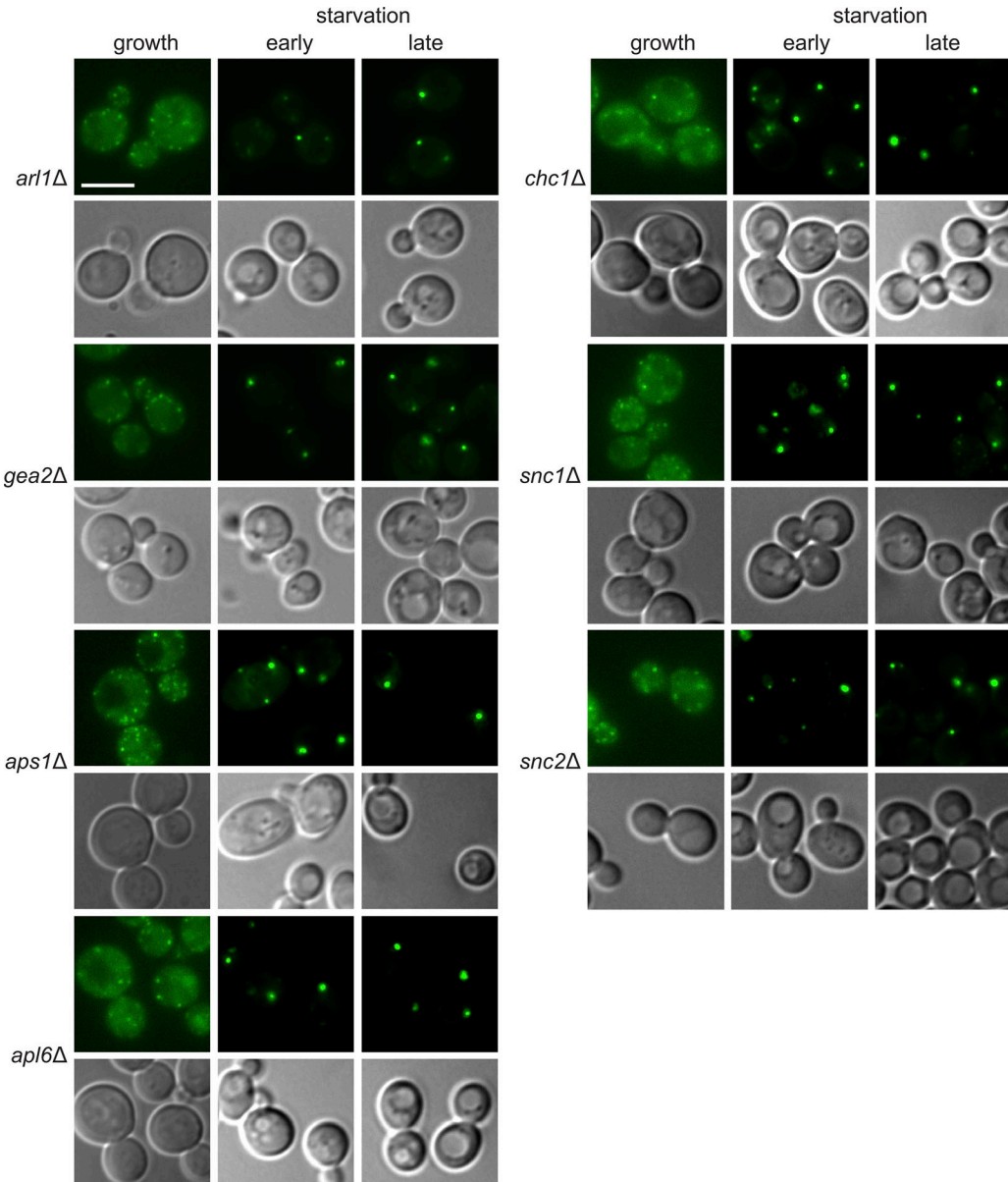

Figure S2. **No CUPS defect in cells lacking Gea2, Arl1, Chc1, Apl6, Aps1, Snc1, or Snc2.** The indicated deletion strains expressing Grh1-2xGFP were grown to the log phase and starved for 2.5 h. Scale bar = 2 µm.

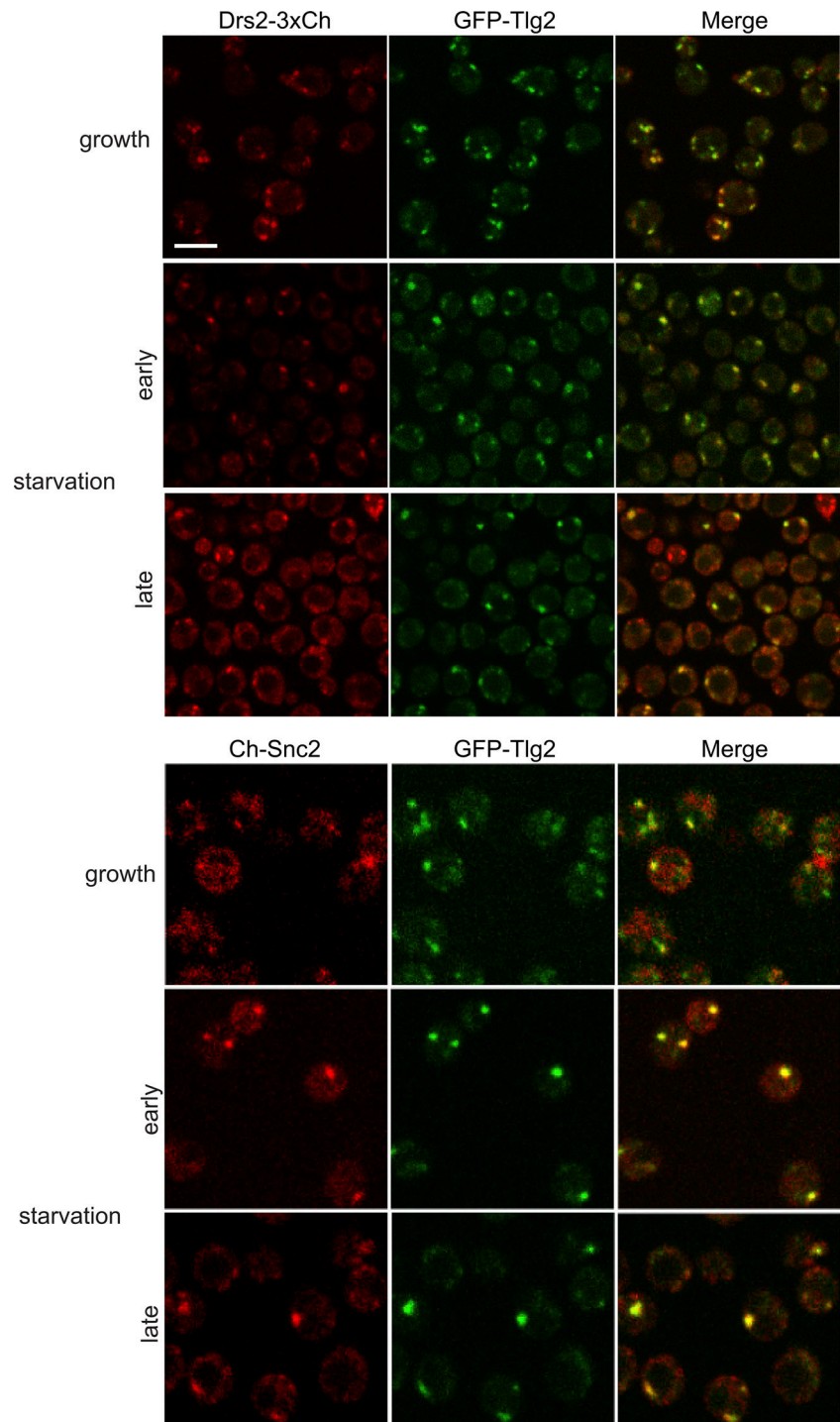

Figure S3.   **Drs2, Tlg2, and Snc2 label the same compartment in starvation—modified TGN. Cells co-expressing GFP-Tlg2 with Drs2-3xmCherry or mCherry-Snc2 were visualized by spinning-disk confocal microscopy in the indicated conditions.** In both combinations, average Pearson's coefficient increased from ∼0.3 in the growth to ∼0.7 in the starvation condition (*n* = 25–40 cells). Scale bar = 2 µm.

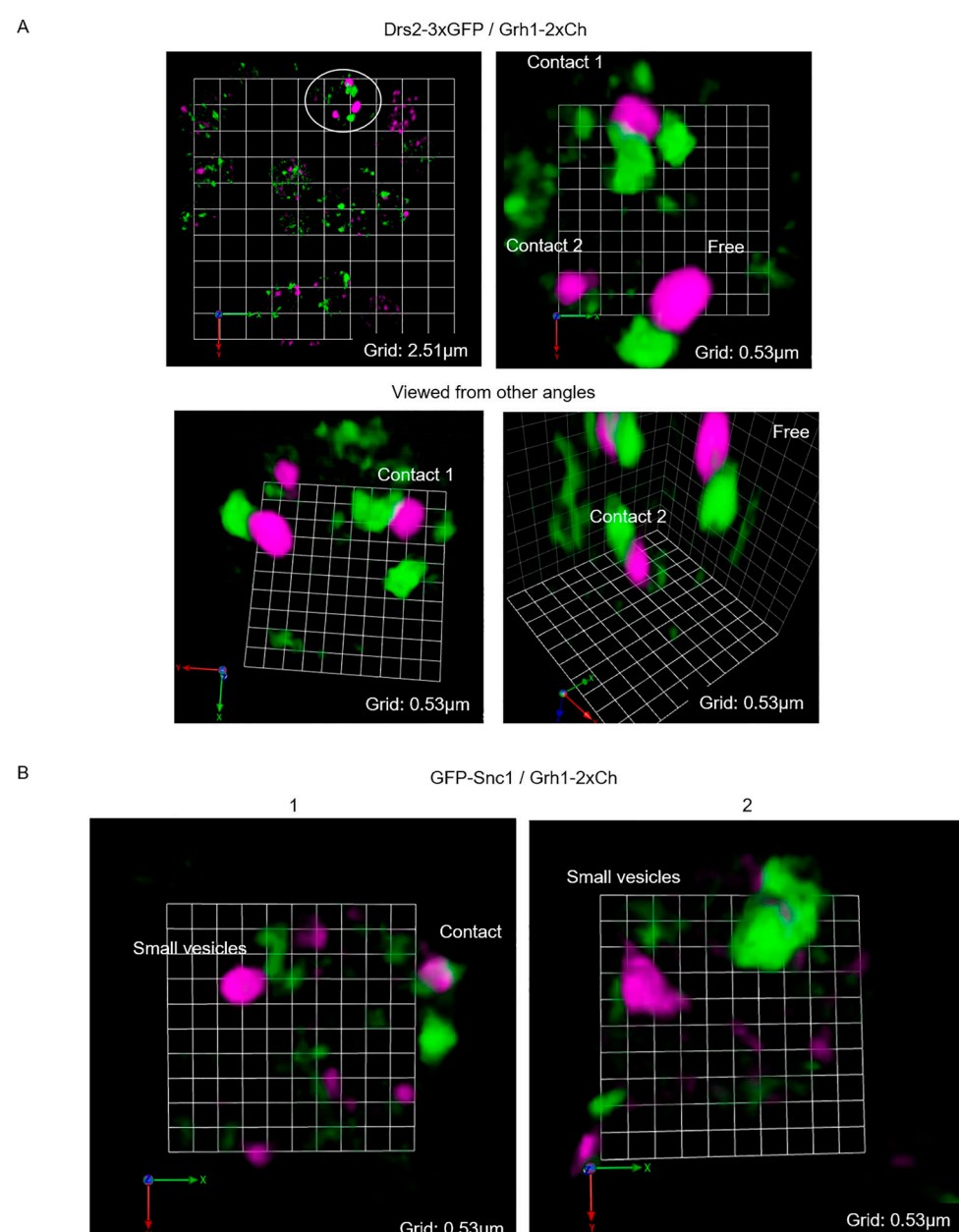

Figure S4. **Drs2 and Snc2 also label small vesicles that contact with or are near CUPS. (A and B)** Grh1-2xCherry (magenta) cells co-expressing either (A) Drs2-3xGFP (green) or (B) GFP-Snc2 (green) were cultured in the starvation condition for 1 h.

Video 1. **Dynamic structure of CUPS revealed by SCLIM.** Cells expressing Grh1-2xGFP were incubated in 2% potassium acetate for 3 h (late starvation condition) and visualized by SCLIM. The movie shows dynamic behavior around Region 1 (moving structures) of Fig. 5 D, with 20-s intervals, played at 60× speed. Grid = 0.82 µm.

Video 2. **Dynamic structure of CUPS revealed by SCLIM.** The same sample of late starvation as in Video 1. Around Region 2 (nonmoving structure) of Fig. 5 D is shown with 20-s intervals, played at 60× speed. Grid = 0.87 µm.

Video 3. **Dynamic structure of CUPS revealed by SCLIM.** The same sample of late starvation as in Video 1. Another region of Fig. 5 D (upper light area) shows dynamic Grh1 structures that occasionally contact one another, with 20-s intervals, played at 60× speed. Grid = 0.82 µm.

Video 4. **Dynamic structure of CUPS revealed by SCLIM.** The same sample of late starvation as in Video 1. Region 2 of Fig. 5 D at around 50 frames shows a cuplike structure of Grh1 that appeared to have formed by fusion and become more stable, with 20-s intervals, played at 60× speed. Grid = 0.25 µm.

Video 5.   **Dynamic structure of CUPS revealed by SCLIM.** Cells expressing Grh1-2mCherry and GFP-Snc2 were incubated in 2% potassium acetate for 1 h (early starvation condition) and visualized by SCLIM. The movie shows potential fusion of the Grh1-positive structure, with 10-s intervals, played at 20× speed. Grid = 0.53 µm.

Video 6.   **Dynamic structure of CUPS revealed by SCLIM.** The same cells as in Video 1, showing potential fission of the Grh1-positive structure, with 10-s intervals, played at 20× speed. Grid = 0.35 µm.

Video 7.   **Dynamic structure of CUPS revealed by SCLIM.** The same cells as in Video 1, showing potential fission of the Grh1-positive structure, with 10-s intervals, played at 20× speed. Grid = 0.53 µm.

Video 8.   **Dynamics of Drs2 and Tlg2 structures in growth.** Cells expressing Drs2-3xCherry (magenta) and GFP-Tlg2 (green) were visualized in the growth condition by SCLIM (Fig. 6 A, lower left panel), with 4-s intervals, played at 20× speed. Grid = 0.41 µm.

Video 9.   **Dynamics of Drs2 and Tlg2 structures in starvation.** Cells expressing Drs2-3xCherry (magenta) and GFP-Tlg2 (green) were visualized at 2-h starvation by SCLIM, with 10-s intervals, played at 20× speed. Grid = 0.54 µm.

Video 10.   **Dynamics of Drs2 and Tlg2 structures in starvation.** Another region of the same sample as in Video 6, with 10-s intervals, played at 20× speed. Grid = 0.54 µm.

Video 11.   **Cells expressing Grh1-2xmCherry (magenta) and GFP-Tlg2 (green) were cultured in the starvation condition for 1.5 h and observed by SCLIM, with 20-s intervals played at 20× speed.**

Video 12.   **Cells expressing Grh1-2xmCherry (magenta) and Drs2-3xGFP (green) were cultured in the starvation condition for 1.5 h and observed by SCLIM, with 10-s intervals, played at 20× speed.**

Video 13.   **Drs2 labels small vesicles that contact with or are near CUPS.** Cells expressing Grh1-2xmCherry and Drs2-3xGFP were cultured in the starvation condition for 1 h and visualized by SCLIM (see Fig. S4 A), with 10-s intervals, played at 20× speed. Grid = 0.53 µm.

Video 14.   **Drs2 labels small vesicles that contact with or are near CUPS.** The same cell as in Video 4, with 10-s intervals, played at 20× speed. Grid = 0.53 µm.

Video 15.   **Drs2 labels small vesicles that contact with or are near CUPS.** The same cell as in Video 4, with 10-s intervals, played at 20× speed. Grid = 0.53 µm.

Video 16.   **Snc2 also labels small vesicles that contact with or are near CUPS.** Cells expressing Grh1-2xmCherry and GFP-Snc2 were cultured in the starvation condition for 1 h and visualized by SCLIM (see Fig. S4 B), with 10-s intervals, played at 20× speed. Grid = 0.53 µm.

Video 17.   **Snc2 also labels small vesicles that contact with or are near CUPS.** The same cell as in Video 4, with 10-s intervals, played at 20× speed. Grid = 0.53 µm.

