## [Peer Review File · The Journal of Cell Biology]

The pathway of unconventional protein secretion involves CUPS and a modified Trans Golgi Network

Amy Curwin, Kazuo Kurokawa, Gonzalo Bigliani, Nathalie Brouwers, Akihiko Nakano, and Vivek Malhotra

Corresponding Author(s): Vivek Malhotra, Centre for Genomic Regulation

Review Timeline:

Submission Date:	2023-12-22
Editorial Decision:	2024-02-01
Revision Received:	2024-12-03
Editorial Decision:	2024-12-16
Revision Received:	2025-01-27

Monitoring Editor: Ira Mellman

Scientific Editor: Andrea Marat

Transaction Report:

DOI: <https://doi.org/10.1083/jcb.202312120>

February 1, 2024

Re: JCB manuscript #202312120

Dr. Vivek Malhotra
CRG, Centre de Regulacio Genomica
quantitative cell biology program, Dr Aiguader 88
Barcelona 08003
Spain

Dear Vivek,

Thank you for submitting your manuscript entitled "CUPS and a modified late Golgi function in unconventional protein secretion". The manuscript was assessed by expert reviewers, whose comments are appended to this letter. We invite you to submit a revision if you can address the reviewers' key concerns, as outlined here.

You will see that the reviewers are positive about the potential interest of your study for the membrane trafficking community. However, while the expert reviewers were able to appreciate the advance, they are concerned that for the general readership of JCB certain aspects need to be more thoroughly explained. They have provided constructive feedback regarding points to clarify, including where additional background information or experimental details are required. In addition, reviewer 2 has suggestions of a few straightforward experiments to attempt to increase the mechanistic details. While the super resolution experiment would be an appreciated addition, it is not necessary. Otherwise, please attempt the other suggested experiments and respond thoroughly to all other reviewer points.

GENERAL GUIDELINES:

Text limits: Character count for an Article is < 40,000, not including spaces. Count includes title page, abstract, introduction, results, discussion, and acknowledgments. Count does not include materials and methods, figure legends, references, tables, or supplemental legends.

Figures: Articles may have up to 10 main text figures. Figures must be prepared according to the policies outlined in our Instructions to Authors, under Data Presentation, <https://jcb.rupress.org/site/misc/ifora.xhtml>. All figures in accepted manuscripts will be screened prior to publication.

*****IMPORTANT:** It is JCB policy that if requested, original data images must be made available. Failure to provide original images upon request will result in unavoidable delays in publication. Please ensure that you have access to all original microscopy and blot data images before submitting your revision. ***

Supplemental information: There are strict limits on the allowable amount of supplemental data. Articles may have up to 5 supplemental figures. Up to 10 supplemental videos or flash animations are allowed. A summary of all supplemental material should appear at the end of the Materials and methods section.

Please note that JCB now requires authors to submit Source Data used to generate figures containing gels and Western blots with all revised manuscripts. This Source Data consists of fully uncropped and unprocessed images for each gel/blot displayed in the main and supplemental figures. Since your paper includes cropped gel and/or blot images, please be sure to provide one Source Data file for each figure that contains gels and/or blots along with your revised manuscript files. File names for Source Data figures should be alphanumeric without any spaces or special characters (i.e., SourceDataF#, where F# refers to the associated main figure number or SourceDataFS# for those associated with Supplementary figures). The lanes of the gels/blots should be labeled as they are in the associated figure, the place where cropping was applied should be marked (with a box), and molecular weight/size standards should be labeled wherever possible.

The typical timeframe for revisions is three to four months. While most universities and institutes have reopened labs and

allowed researchers to begin working at nearly pre-pandemic levels, we at JCB realize that the lingering effects of the COVID-19 pandemic may still be impacting some aspects of your work, including the acquisition of equipment and reagents. Therefore, if you anticipate any difficulties in meeting this aforementioned revision time limit, please contact us and we can work with you to find an appropriate time frame for resubmission. Please note that papers are generally considered through only one revision cycle, so any revised manuscript will likely be either accepted or rejected.

Thank you for this interesting contribution to Journal of Cell Biology. You can contact us at the journal office with any questions at cellbio@rockefeller.edu.

Sincerely,

Ira Mellman, Ph.D.
Editor

Andrea L. Marat, Ph.D.
Senior Scientific Editor

Journal of Cell Biology

Reviewer #1 (Comments to the Authors (Required)):

This manuscript seeks to add to our understanding of an unconventional secretory pathway with focus on yeast (please state that in the results). Previously the authors showed that the pathway is starvation induced and requires a protein called Grh1 in yeast (or GORASP in mammals), and PI4P. Here they report a possible role for the Drs2 P-type ATPase/lipid flippase and its partner Rcy1. They conclude that the process involves a modified TGN that contains Drs2, Tlg2 t-SNARE and Snc2 v-SNARE.

The high resolution videos of these compartments are remarkable and worthy of presentation in JCB. However the manuscript is very poorly written and needs significant editing for grammar, clarity and precision related to the conclusions.

1. Please explain what Drs2 is, with proper citations on pages 4 and 5. Also, Pazos et al. 2023 showed that Drs2 plays a role in cytosol to vacuole transport. Please clarify if this is the same process used for Acb1 secretion.
2. Page 5. "The foci of Grh1 and Drs2 were never observed to be stably localized". What does this mean? Stably co-localized? "Diffuse vesicular staining in the cytoplasm" please remind the reader that Drs2 has 10 transmembrane domains so must reflect membrane localization.
3. Please do a better job of introducing Rcy1. This is especially important if we are supposed to believe that Rcy1 deletion strains phenocopy loss of Drs2 with regard to unconventional secretion (page 8). Do they?
4. Page 9. "Most analyses of v-SNARE itinerary" please correct
Not clear relationship between Graham's mNG-Snc1 and what the authors do--why is it relevant to mention here? (page 9)
5. Page 11. "CUPS form by dynamic interactions between Grh1 containing membranes" which movie showed formation? They all seem to show moving vesicles alone or attached to another marker. PLEASE label all movies in terms of which marker is which color and which condition.
6. The yeast reviewers will be better able to judge the novelty of the SNARE gymnastics as the TGN and endosomes are more complex-seeming than in mammalian cells; please make this clearer for the reader.
7. Figure 8 should be clear to the reader without a legend. How do we know that CUPS derives from early Golgi and given the rarity of seeing various structures in the present study, how do we know that they are relevant intermediates? A figure showing where the various new SNARES are and how this relates to CVT and how secretion happens would be much more helpful here.

Conclusion: The authors need to do a better job of summarizing what is new here, why it is important, and what happens after this compartment forms to enable secretion. Also, what is likely but not yet shown. With careful revision and rewriting, the article may be a wonderful contribution to JCB.

Reviewer #2 (Comments to the Authors (Required)):

The work by Curwin and colleagues highlights roles for Drs2, Rcy1, and the SNAREs Snc1/2 in CUPS formation, facilitating

unconventional secretion of proteins that lack a signal sequence in yeast. Additionally, the authors use super resolution confocal imaging to examine the formation of CUPS for the first time. They also suggest that the TGN is consumed during starvation, but the resulting carriers generated do not fuse to CUPS. Although the fate of these carriers remains unclear, the study nicely describes several factors necessary for CUPS formation. Some additional mechanistic insights seem necessary to support publication, however.

1. The authors claim an interaction between Drs2 and Rcy1 is necessary for CUPS formation. Mutations in either Drs2 or Rcy1 that specifically disrupt their interaction seem necessary to make this claim.
2. Is Drs2 flippase activity necessary for CUPS formation? A mutant that specifically disrupts Drs2 flippase activity should be tested.
3. Super resolution imaging should be used to study what occurs to Grh1 in the absence of Drs2, Rcy1, and Snc1/2. If Grh1 dynamics is similar in all cases, it would support the existence of the CUPS formation pathway the authors postulate and strengthen the manuscript.
4. Electron microscopy of cells lacking Drs2, Rcy1, and/or Snc1/2 would be helpful to understand what happens to the TGN and CUPS during starvation.

Reviewer #3 (Comments to the Authors (Required)):

Synopsis: The authors extend on their previous studies on the biogenesis of CUPS - a compartment that forms upon starvation, and which serves to secrete a number of cytoplasmic proteins that lack conventional signals for secretion. Using a combination of genetics, biochemistry and sophisticated cell biological methods the authors reveal a role for several proteins in the biogenesis of CUPS which function at the interface of the endosomal / TGN in yeast cells. The authors' data is consistent with their submission that the formation and incorporation of fragmented early TGN compartment into CUPS plays a critical role in unconventional secretion. On the whole is this a carefully executed and convincing study, that at least for as far as it goes, will be of significant interest to members of the Cell Biology community with an interest in membrane trafficking and unconventional secretion.

Critic:

Page 10 "None of these v-SNARE proteins could be co-localized with Grh1 in growth conditions." Presumably the authors intended SNAREs rather than v-SNAREs? As Tlg2 is a t-SNARE, and this SNARE protein also does not co-localize with Grh1.

Page 19. "For preparation of cell wall extracts for mass spectrometry analysis, no BSA carrier protein was added and the proteins were precipitated with acetone and not TCA." As far as I can tell there is no proteomic data presented in this manuscript. Delete as not relevant?

Figure 1 and 2: I can find no description is provided for how the authors determine whether CUPS are present or the degree of fragmentation of CUPS. The data in Figure shows Grh1 puncta in the various mutants and in WT cells. Whilst of phenotype of WT cells upon starvation is quite obvious, it is somewhat more indeterminate for the mutants examined. What criteria were used to exclude the occasional larger puncta apparent for cells lacking DCR2 (in the early panel) from being CUPS? The same question applies to the data presented in Figure 2.

Figure 3: What accounts for the apparent differences in the amount of Sod1 detected in wildtype cells in panels A and B? Far less protein is apparent in panel B for Sod1. I note from the methods section that protein precipitation involves the addition of BSA to each sample. Do the / did the authors look at precipitation of BSA with TCA for the purpose of adjusting the relative amounts of protein loaded for each sample?

Figure 4B has no scale bar

Figure 7 A and B: It is not immediately obvious what object(s) are being highlighted here. Perhaps the use of an arrow would be more definitive than using a triangle?

Figure 8 In this reviewer's opinion the legend for this figure would benefit from (as would the Discussion) indicating relevant findings from previously published work (or what is yet to be determined) that accounts for some of the proposed steps in the biogenesis and later consumption of CUPS. (e.g., biogenesis of CUPS apparently does not involve COPI, but consumption / subsequent fusion with the ER does). Presumably SNAREs are required for fusion of CUPS with ER as well as fusion of vesicles containing non-conventionally secreted (and formerly cytoplasmic proteins). Similarly, fragmentation / vesiculation of the TGN does not explain how cytoplasmic proteins are incorporated in presumptive vesicles that fuse with plasma membrane. Some discussion of known vs outstanding questions would help to further engage future readers.

Figure S3 - no grid size is indicated

Dear Editors,

We would like to express our gratitude to the reviewers for their insightful feedback and constructive critiques. In response to the letter from the monitoring editor, Dr. Mellman, we have addressed the following two major concerns:

1. **Demonstrating the Requirement of Drs2 Activity for CUPS Formation:** This issue required significant time to resolve, primarily due to the tendency of Drs2 mutants to revert, which complicated our ability to express mutants with defective functions for testing in the unconventional secretion pathway. Despite employing rapid degradation techniques, the CRISPR knock-ins necessitated the selection of mutants, during which revertants emerged. Nonetheless, we have now successfully demonstrated that the enzymatic activity of Drs2 is essential for CUPS formation (new Figure S1). Although we have not yet explored whether mutations in the Drs2 binding site for Rcy1 affect UPS, as this presents technical challenges, we have confirmed that only the Rcy1 pathway among the various Drs2 pathways is essential for UPS.
2. **Improving Clarity and Addressing Missing References:** In response to the feedback, I have rewritten the manuscript to enhance clarity. The original presentation was inadequate and did not effectively convey the significance of our data. The revised version now offers a more coherent narrative, clearly articulating the roles of CUPS and the modified TGN as compartments responsible for the collection and sorting of signal-sequence-lacking proteins. This is now explicitly stated in both the abstract and the discussion sections.
3. **Loss of Co-Author Kazuo Kurokawa:** It is with deep sadness that we note the passing of our co-author, Kazuo Kurokawa, after his courageous battle with pancreatic cancer. We had hoped to submit the paper before his passing, but regrettably, we were unable to do so.

With these revisions, we believe the manuscript is now better suited for publication in the *Journal of Cell Biology*.

Our response, in italics, to the reviewers' comments follows.

Reviewer #1 (Comments to the Authors (Required)):

This manuscript seeks to add to our understanding of an unconventional secretory pathway with focus on yeast (please state that in the results). Previously the authors showed that the pathway is starvation induced and requires a protein called Grh1 in yeast (or GORASP in mammals), and PI4P. Here they report a possible role for the Drs2 P-type ATPase/lipid flippase and its partner Rcy1. They conclude that the process involves a modified TGN that contains Drs2, Tlg2 t-SNARE and Snc2 v-SNARE.

The high-resolution videos of these compartments are remarkable and worthy of presentation in

JCB. However, the manuscript is very poorly written and needs significant editing for grammar, clarity and precision related to the conclusions.

We have rewritten nearly the entire paper to enhance its readability and clarify the meaning of our data. We would like to thank the reviewer for emphasizing the need for clarification and for encouraging us to improve the quality of our writing.

1. Please explain what Drs2 is, with proper citations on pages 4 and 5. Also, Pazos et al. 2023 showed that Drs2 plays a role in cytosol to vacuole transport. Please clarify if this is the same process used for Acb1 secretion.

We explained Drs2 in the introduction with citations. However, we have not tested the involvement of Drs2 in the transport from the cytosol to the vacuole. There is no reason to invoke this pathway for Acb1 secretion, as we have already demonstrated that it is not involved in the secretion of Acb1 (Duran et al., JCB 2010). Additionally, we have never observed Acb1 in the yeast vacuole.

2. Page 5. "The foci of Grh1 and Drs2 were never observed to be stably localized".

What does this mean? Stably co-localized? "Diffuse vesicular staining in the cytoplasm" please remind the reader that Drs2 has 10 transmembrane domains so must reflect membrane localization.

This is now changed to "Curiously, Drs2, a transmembrane protein, also re-localized to 1-3 larger foci per cell in addition to a diffuse vesicular staining in the cytoplasm (Figure 1A)."

3. Please do a better job of introducing Rcy1. This is especially important if we are supposed to believe that Rcy1 deletion strains phenocopy loss of Drs2 with regard to unconventional secretion (page 8). Do they?

This is now explained in the text "Rcy1, an F-box containing protein binds Drs2 to activate the latter for the retrograde recycling of exocytic v-SNAREs Snc1 to the TGN. This interaction is also via the C-terminal domain of Drs2 in a region proximal to the PI4P binding site and partially overlapping with the Gea2 binding site (Furuta, Fujimura-Kamada et al. 2007, Hanamatsu, Fujimura-Kamada et al. 2014). This function of Rcy1 is independent of the cullin-Ub conjugating E2 ligase (Cdc34) pathway (Galan et al., 2001).

We cannot test Drs2 in the unconventional secretion pathway because it affects many cellular processes including membrane integrity. But we have shown the Drs2 is required for CUPS formation. So, we can visualize effects on CUPS but cannot measure the effect on secretion of Acb1 and SOD1. Rcy1, on the other hand can be used to measure both CUPS formation and effects on secretion. Both CUPS formation and Acb1 secretion is affected in RCY1 deleted cells.

4. Page 9. "Most analyses of v-SNARE itinerary" please correct

In the revised version, this is now on page 10. The statement is Changed to "Most analyses of v-SNARE location and trafficking have been conducted using overexpressed N-terminal GFP-tagged Snc1, which, at steady state, primarily labels the plasma membrane of growing buds and some internal structures.

Not clear relationship between Graham's mNG-Snc1 and what the authors do--why is it relevant to mention here? (page 9)

The point is that analyses of v-SNAREs have always been conducted under overexpression conditions, whereas Graham's mNG version showed lower, more endogenous-like expression levels exhibited different localization. The text now states "However, Graham and colleagues developed an mNG-Snc1 construct able to be visualized at much lower expression levels, which preferentially labeled the TGN and endosomes, indicating that overexpression of these SNAREs leads to aberrant localization in steady state conditions (Best, Xu et al. 2020). To avoid plasmid overexpression altogether, we integrated the GFP tag at the N-terminus of each SNARE at its endogenous locus under the control of the Sed5 promoter.

5. Page 11. "CUPS form by dynamic interactions between Grh1 containing membranes" which movie showed formation?

Now on page 12: "Overall, the SCLIM analyses reveal that CUPS form through dynamic interactions among Grh1-containing membranes, likely involving both fusion and fission" (Movies 1–3).

6. The yeast reviewers will be better able to judge the novelty of the SNARE gymnastics as the TGN and endosomes are more complex-seeming than in mammalian cells; please make this clearer for the reader.

Done

7. Figure 8 should be clear to the reader without a legend. How do we know that CUPS derives from early Golgi and given the rarity of seeing various structures in the present study, how do we know that they are relevant intermediates?

We have shown this before in Bruns et al., 2011. They form under conditions of unconventional secretion and contain Acb1 (Bruns et al., 2011)

A figure showing where the various new SNAREs are and how this relates to CVT and how secretion happens would be much more helpful here.

As stated above, our pathway is not related to the CVT pathway for delivering cytoplasmic proteins to the vacuole. We have now discussed the possible mechanisms of cargo sorting and export through the combined involvement of CUPS and the modified TGN in the discussion section. We are also excited by the new report that claims mature interleukin 1 β enters trans-Golgi network in mammalian cells for its export by the cells (Caielli et al., 2024). This nicely aligns with the data presented here.

Conclusion: The authors need to do a better job of summarizing what is new here, why it is important, and what happens after this compartment forms to enable secretion. Also, what is likely but not yet shown. With careful revision and rewriting, the article may be a wonderful contribution to JCB.

Thanks. We have almost rewritten the entire paper to address reviewers concerns.

Reviewer #2 (Comments to the Authors (Required)):

The work by Curwin and colleagues highlights roles for Drs2, Rcy1, and the SNAREs Snc1/2 in CUPS formation, facilitating unconventional secretion of proteins that lack a signal sequence in yeast. Additionally, the authors use super resolution confocal imaging to examine the formation of CUPS for the first time. They also suggest that the TGN is consumed during starvation, but the resulting carriers generated do not fuse to CUPS. Although the fate of these carriers remains unclear, the study nicely describes several factors necessary for CUPS formation. Some additional mechanistic insights seem necessary to support publication, however.

1. The authors claim an interaction between Drs2 and Rcy1 is necessary for CUPS formation. Mutations in either Drs2 or Rcy1 that specifically disrupt their interaction seem necessary to make this claim.

This is extremely difficult as explained in the general comments to the editor. This is primarily due to the reversion of Drs2 mutants, complicating our ability to express mutants with defective functions for testing in the unconventional secretion pathway. Despite employing rapid degradation techniques, CRISPR knock-ins necessitate the selection of mutants, during which revertants began to emerge. Nevertheless, we have successfully demonstrated that the enzymatic activity of Drs2 is essential for CUPS formation (new Figure S1). We have not yet explored whether mutations in the Drs2 binding site for Rcy1 affect CUPS, as this presents technical challenges. However, we have established that only the Rcy1 pathway among the various pathways of Drs2 is requisite for UPS.

2. Is Drs2 flippase activity necessary for CUPS formation? A mutant that specifically disrupts Drs2 flippase activity should be tested.

We now report that the enzymatic activity of Drs2 is required for CUPS formation (new Figure S1).

3. Super resolution imaging should be used to study what occurs to Grh1 in the absence of Drs2, Rcy1, and Snc1/2. If Grh1 dynamics is similar in all cases, it would support the existence of the CUPS formation pathway the authors postulate and strengthen the manuscript.

This is currently not possible because of suppression problems in deletion mutants. Acute depletion is technically very challenging because they are all membrane proteins. Also, the editor stated that this experiment was not necessary

4. Electron microscopy of cells lacking Drs2, Rcy1, and/or Snc1/2 would be helpful to understand what happens to the TGN and CUPS during starvation.

This is beyond the scope at present. Also, the editor stated that this was not necessary to support our proposal.

Reviewer #3 (Comments to the Authors (Required)):

Synopsis: The authors extend on their previous studies on the biogenesis of CUPS - a compartment that forms upon starvation, and which serves to secrete a number of cytoplasmic proteins that lack conventional signals for secretion. Using a combination of genetics, biochemistry and sophisticated cell biological methods the authors reveal a role for several proteins in the biogenesis of CUPS which function at the interface of the endosomal / TGN in yeast cells. The authors' data is consistent with their submission that the formation and incorporation of fragmented early TGN compartment into CUPS plays a critical role in unconventional secretion. On the whole is this a carefully executed and convincing study, that at least for as far as it goes, will be of significant interest to members of the Cell Biology community with an interest in membrane trafficking and unconventional secretion.

Critic:

Page 10 "None of these v-SNARE proteins could be co-localized with Grh1 in growth conditions." Presumably the authors intended SNAREs rather than v-SNAREs? As Tlg2 is a t-SNARE, and this SNARE protein also does not co-localize with Grh1.

Yes, should say SNARE. Corrected now on page 11.

Page 19. "For preparation of cell wall extracts for mass spectrometry analysis, no BSA carrier protein was added and the proteins were precipitated with acetone and not TCA." As far as I can tell there is no proteomic data presented in this manuscript. Delete as not relevant?

Corrected.

Figure 1 and 2: I can find no description is provided for how the authors determine whether CUPs are present or the degree of fragmentation of CUPs. The data in Figure shows Grh1 puncta in the various mutants and in WT cells. Whilst of phenotype of WT cells upon starvation is quite obvious, it is somewhat more indeterminant for the mutants examined. What criteria were used to exclude the occasional larger puncta apparent for cells lacking DCR2 (in the early panel) from being CUPs? The same question applies to the data presented in Figure 2.

We only note the phenotype looks different. There is no criteria to say the large puncta is CUPS or not, per se, but early time are not CUPS even in WT cells. This has been shown previously, and in this paper, that immature CUPS look different (at EM level). CUPS formation is dynamic and takes some time and at later times, in mutant cells, the absence of large foci is much more obvious.

Figure 3: What accounts for the apparent differences in the amount of Sod1 detected in wildtype cells in panels A and B? Far less protein is apparent in panel B for Sod1. I note from the methods section that protein precipitation involves the addition of BSA to each sample. Do the / did the authors look at precipitation of BSA with TCA for the purpose of adjusting the relative amounts of protein loaded for each sample?

The same BSA is added to all samples to aid in the precipitation process. This is standard for low total protein samples. It is not for normalizing and amounts loaded were not adjusted in any way.

Figure 4B has no scale bar

Corrected

Figure 7 A and B: It is not immediately obvious what object(s) are being highlighted here. Perhaps the use of an arrow would be more definitive than using a triangle?

This has been corrected and explained in the legend.

Figure 8 In this reviewer's opinion the legend for this figure would benefit from (as would the Discussion) indicating relevant findings from previously published work (or what is yet to be determined) that accounts for some of the proposed steps in the biogenesis and later consumption of CUPs. (e.g., biogenesis of CUPs apparently does not involve COPI, but consumption / subsequent fusion with the ER does). Presumably SNAREs are required for fusion of CUPs with ER as well as fusion of vesicles containing non-conventionally secreted (and formerly

cytoplasmic proteins). Similarly, fragmentation / vesiculation of the TGN does not explain how cytoplasmic proteins are incorporated in presumptive vesicles that fuse with plasma membrane. Some discussion of known vs outstanding questions would help to further engage future readers.

I have changed this in discussion to state our working hypothesis more clearly.

Figure S3 - no grid size is indicated

Corrected

December 16, 2024

RE: JCB Manuscript #202312120R

Vivek Malhotra
CRG, Centre de Regulacio Genomica

Dear Vivek,

Thank you for submitting your revised manuscript entitled "The pathway of unconventional protein secretion involves CUPS and a modified Trans Golgi Network". The reviewers now support publication so we would be happy to publish your paper in JCB pending final revisions necessary to meet our formatting guidelines (see details below). In your final revision, please be sure to thoroughly address the reviewers' final comments.

A. MANUSCRIPT ORGANIZATION AND FORMATTING:

- 1) Text limits: Character count for Articles is < 40,000, not including spaces. Count includes abstract, introduction, results, discussion, and acknowledgments. Count does not include title page, figure legends, materials and methods, references, tables, or supplemental legends.
- 2) Figures limits: Articles may have up to 10 main text figures.
- 3) * Figure formatting: Scale bars must be present on all microscopy images, including inset magnifications. *Molecular weight or nucleic acid size markers must be included on all gel electrophoresis.* Aspect ratios of images may not be altered. In order to accommodate readers with red-green color blindness, we ask that you please change all red/green color schemes.
- 4) Statistical analysis: Error bars on graphic representations of numerical data must be clearly described in the figure legend. The number of independent data points (n) represented in a graph must be indicated in the legend. Statistical methods should be explained in full in the materials and methods. For figures presenting pooled data the statistical measure should be defined in the figure legends. Please also be sure to indicate the statistical tests used in each of your experiments (either in the figure legend itself or in a separate methods section) as well as the parameters of the test (for example, if you ran a t-test, please indicate if it was one- or two-sided, etc.). Also, if you used parametric tests, please indicate if the data distribution was tested for normality (and if so, how). If not, you must state something to the effect that "Data distribution was assumed to be normal but this was not formally tested."
- 5) Abstract and title: The abstract should be no longer than 160 words and should communicate the significance of the paper for a general audience. The title should be less than 100 characters including spaces. Make the title concise but accessible to a general readership.
- 6) Materials and methods: Should be comprehensive and not simply reference a previous publication for details on how an experiment was performed. Please provide full descriptions in the text for readers who may not have access to referenced manuscripts.
- 7) ** All antibodies, cell lines, animals, and tools used in the manuscript should be described in full, including accession numbers for materials available in a public repository such as the Resource Identification Portal. ** Please be sure to provide the sequences for all of your primers/oligos and RNAi constructs in the materials and methods. You must also indicate in the methods the source, species, and catalog numbers (where appropriate) for all of your antibodies. Please also indicate the acquisition and quantification methods for immunoblotting/western blots.
- 8) Microscope image acquisition: The following information must be provided about the acquisition and processing of images:
 - a. Make and model of microscope
 - b. Type, magnification, and numerical aperture of the objective lenses
 - c. Temperature
 - d. Imaging medium
 - e. Fluorochromes
 - f. Camera make and model
 - g. Acquisition software

h. Any software used for image processing subsequent to data acquisition. Please include details and types of operations involved (e.g., type of deconvolution, 3D reconstitutions, surface or volume rendering, gamma adjustments, etc.).

10) Supplemental materials: There are strict limits on the allowable amount of supplemental data. Articles may have up to 5 supplemental figures. Please also note that tables, like figures, should be provided as individual, editable files. A summary of all supplemental material should appear at the end of the Materials and methods section.

13) ORCID IDs: ORCID IDs are unique identifiers allowing researchers to create a record of their various scholarly contributions in a single place. Please note that ORCID IDs are now *required* for all authors. At resubmission of your final files, please be sure to provide your ORCID ID and those of all co-authors.

Please note that JCB now requires authors to submit Source Data used to generate figures containing gels and Western blots with all revised manuscripts. This Source Data consists of fully uncropped and unprocessed images for each gel/blot displayed in the main and supplemental figures. Since your paper includes cropped gel and/or blot images, please be sure to provide one Source Data file for each figure that contains gels and/or blots along with your revised manuscript files. File names for Source Data figures should be alphanumeric without any spaces or special characters (i.e., SourceDataF#, where F# refers to the associated main figure number or SourceDataFS# for those associated with Supplementary figures). The lanes of the gels/blots should be labeled as they are in the associated figure, the place where cropping was applied should be marked (with a box), and molecular weight/size standards should be labeled wherever possible. Source Data files will be made available to reviewers during evaluation of revised manuscripts and, if your paper is eventually published in JCB, the files will be directly linked to specific figures in the published article.

Journal of Cell Biology now requires a data availability statement for all research article submissions. These statements will be published in the article directly above the Acknowledgments. The statement should address all data underlying the research presented in the manuscript. Please visit the JCB instructions for authors for guidelines and examples of statements at (<https://rupress.org/jcb/pages/editorial-policies#data-availability-statement>).

B. FINAL FILES:

****It is JCB policy that if requested, original data images must be made available to the editors. Failure to provide original images upon request will result in unavoidable delays in publication. Please ensure that you have access to all original data images prior to final submission.****

****The license to publish form must be signed before your manuscript can be sent to production. A link to the electronic license to publish form will be sent to the corresponding author only. Please take a moment to check your funder requirements before choosing the appropriate license.****

Thank you for your attention to these final processing requirements. Please revise and format the manuscript and upload materials within 7 days. If you need an extension for whatever reason, please let us know and we can work with you to determine a suitable revision period.

Thank you for this interesting contribution, we look forward to publishing your paper in Journal of Cell Biology.

Sincerely,

Ira Mellman, Ph.D.
Editor

Andrea L. Marat, Ph.D.
Deputy Editor

Journal of Cell Biology

Reviewer #1 (Comments to the Authors (Required)):

The revised manuscript is significantly improved and with only textual edits, the story should be accepted for publication.

1. The abstract overstates the data.

"CUPS, a compartment for secretion of signal sequence-lacking proteins, forms through COPI-independent extraction of membranes from early Golgi cisternae,[NOT SHOWN--please be precise] ...

...Notably, while CUPS remain stable, the modified TGN undergoes vesiculation [REMODELING IS MORE ACCURATE?] during the later stages of unconventional secretion.

2. Line 34. "How proteins...remains a fascinating question."

Line 52 add FROM YEAST

3. Line 94. is a lipid flippase and PI4P effector

4. Line 136. and the Drs2 PI4P sensor

5. Fig. S2 is out of focus

6. Line 174. What if SNARE loss makes weird membranes and phagosomes fuse with CUPS? it may not reflect "acquire membranes from..."

7. Line 193. Say which v-SNARE double mutant

8. Line 223. Why does this necessarily indicate that the disappearing SNARE is the one preferred for exocytosis?

9. Line 238. What should the reader conclude here, and how does this match the subheading on line 199? Please be precise

10. Line 300. a "large"? not really large

11. Is it clear that cargo goes into the TGN rather than being wrapped around it? Please state clearly that how cargo engages this pathway remains a major mystery or?

12. Is the title truly accurate? would it not be more precise to say TGN-localized components rather than a modified TGN which is not really shown here?

Reviewer #2 (Comments to the Authors (Required)):

The authors have nicely demonstrated that Drs2 flippase activity is necessary for CUPS formation, which significantly enhances the strength of the manuscript. Since they were unable to test whether a direct interaction between Drs2 and Rcy1 is necessary due to technical challenges, I would suggest they softened language around a specific requirement. In particular, the term 'specifically' in line 24 (Abstract) could be replaced with 'likely' or a similar term. Otherwise, this manuscript appears ready for publication.

Dear Editors,

We thank the reviewers for helping us improve the quality of the paper. The last set of minor changes requested by reviewer #1 are now addressed in the revised manuscript.

Our response to reviewer 1's concerns follow in italics.

The revised manuscript is significantly improved and with only textual edits, the story should be accepted for publication.

Thanks .

1. The abstract overstates the data.

"CUPS, a compartment for secretion of signal sequence-lacking proteins, forms through COPI-independent extraction of membranes from early Golgi cisternae,[NOT SHOWN--please be precise]

We showed this previously that COPI proteins are not required for CUPS formation. This is the correct definition.

...Notably, while CUPS remain stable, the modified TGN undergoes vesiculation [REMODELING IS MORE ACCURATE?] during the later stages of unconventional secretion.

Done

2. Line 34. "How proteins...remains a fascinating question."

We have changed it to fascinating question.

Line 52 add FROM YEAST

Ok.Done

3. Line 94. is a lipid flippase and PI4P effector

Ok. done

4. Line 136. and the Drs2 PI4P sensor

Ok. Done.

5. Fig. S2 is out of focus .

The problem exported file was solved.

6. Line 174. What if SNARE loss makes weird membranes and phagosomes fuse with CUPS? it may not reflect "acquire membranes from..."

This is adding yet another layer of complexity. We do not know anything about the relationship of phagosomes and/or their fusion with CUPS> We respectfully request that we leave the text as is.

7. Line 193. Say which v-SNARE double mutant

OK. Changed it to Snc1 and Snc2 double mutant.

8. Line 223. Why does this necessarily indicate that the disappearing SNARE is the one preferred for exocytosis?

OK. We have now removed the statement , " indicating that Snc1 is the preferred SNARE for exocytosis.

9. Line 238. What should the reader conclude here, and how does this match the subheading on line 199? Please be precise

This is splitting hairs. All this means is that Snc2 and Tlg2 colocalise in the modified TGN that contains Drs2. We do not need to change this.

10. Line 300. a "large"? not really large

Ok. Done

11. Is it clear that cargo goes into the TGN rather than being wrapped around it? Please state clearly that how cargo engages this pathway remains a major mystery or?

This is not at all clear and we respectfully request that we are allowed to leave the statement as is.

12. Is the title truly accurate? would it not be more precise to say TGN-localized components rather than a modified TGN which is not really shown here?

The title is accurate. We see that the compartment contains Drs2, Snc2 and Tlg2. Otherwise we have to invoke another compartment that somehow receives a subset of TGN derived proteins. We should leave it as is.

Sincerely yours,

Vivek Malhotra